# Hundreds of grocery outlets needed across the United States to achieve walkable cities

Drew Horton[1], Tom M. Logan [2] ✉, Emily Speakman [1] & Daphne Skipper[3]

The location of amenities in urban areas fundamentally shapes both sustainability and equity outcomes. While cities worldwide are pursuing walkable neighborhood initiatives, the practical implications of retrofitting existing urban areas remain unclear. How many new facilities are needed, and where should they be located to ensure equitable access? We analyzed supermarket access across 500 U.S. cities using an optimization approach that minimizes both average distance and inequality in its distribution. Unlike traditional methods that focus solely on reducing inequality or average distance, our approach identifies solutions that improve overall accessibility while reducing disparities. We found that 25% of cities could achieve 15-minute walking access by adding five or fewer stores in optimal locations, while more ambitious 5-minute targets would require over 100 additional stores in most cities. These findings demonstrate both the potential for strategic interventions to efficiently improve access and the substantial challenge posed by car-oriented urban development. By identifying priority areas for new facilities while considering distributional impacts, our method can inform multiple stakeholders working to create more sustainable and equitable cities - from local governments using zoning and incentives to state agencies developing funding programs and community organizations advocating for improved food access.

The location of amenities in urban areas fundamentally shapes both sustainability and equity outcomes. While walkable access to daily needs offers multiple benefits—including improved public health[1–3], emissions reductions[4], and stronger social cohesion[5,6]—many cities' car-oriented urban design limits these opportunities[7,8]. Between 1990 and 2014, urban sprawl increased globally by 95%[9], creating environments where essential services like grocery stores remain inaccessible without a car. These design choices disproportionately affect minority and lower-income communities[10,11], compounding existing disparities in health outcomes and quality of life[12,13]. Food access is critical to this challenge—despite being essential for wellbeing, grocery stores in many US cities remain car-dependent and inequitably distributed, with minority and relatively low-income communities facing systematically greater barriers to access[10,11].

Cities worldwide are now articulating visions of improving their residents' proximity to amenities and thereby capitalizing on the benefits of active transport[14]. These range from ambitious 5-min targets in Copenhagen to 20-min goals in cities like Portland and Glasgow, with many cities, including Paris, Ottawa, and Shanghai, adopting 15-minute targets. While car access has expanded mobility for many, true transport freedom requires that walking remains viable for daily needs. This is particularly important for grocery shopping—research in the Netherlands found that 50% of respondents consider 500 m (approximately a 5–6-min walk) too far to carry groceries[15]. This sustainable transition raises critical questions for our existing cities about how they can direct this retrofit efficiently and effectively.

The inequitable distribution of food retailers exemplifies broader patterns of environmental injustice, where disadvantaged populations systematically face greater barriers to accessing essential services[16–18].

[1]Department of Mathematical and Statistical Sciences, University of Colorado, Denver, CO, USA. [2]Department of Civil and Natural Resources Engineering, University of Canterbury, Christchurch, New Zealand. [3]Annapolis, MD, USA. ✉e-mail: tom.logan@canterbury.ac.nz

How can we measure inequality in communities?
The EDE helps us evaluate how evenly resources and burdens are distributed across a population. The EDE represents the average experience of a resident plus a penalty for inequality.

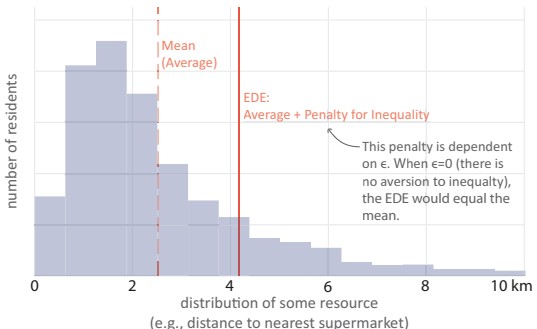

**Fig. 1 | Measuring inequality using the equally distributed equivalent (EDE).** This figure illustrates how the EDE captures both average access and inequality in a community. The gap between the mean (average) distance and the EDE reflects the presence of residents with substantially worse access than average. A larger gap indicates greater inequality in the distribution of access. The size of this gap increases with the inequality aversion parameter, allowing planners to place greater emphasis on improving access for the most disadvantaged residents.

These disparities reflect historical patterns of disinvestment and discriminatory policies, with supermarkets often concentrated in wealthier, predominantly white suburbs. Rather than being naturally occurring phenomena, these spatial patterns result from deliberate policy choices that have created and reinforced inequitable food environments[19]. The health implications are substantial—limited access to healthy food is associated with higher rates of chronic disease[10,20], with disproportionate impacts on racial/ethnic minority and relatively low-income communities[21,22]. The Covid-19 pandemic further exposed these vulnerabilities, with increased food insecurity particularly affecting Black and American Indian populations[23] and relatively low-income households with children experiencing a 22% increase in food insecurity from 2019 to 2020[24].

Distributional justice—how benefits and burdens should be distributed across society[25]—is central to this challenge. While geographic proximity represents just one dimension of food access, improving its distribution remains crucial for enabling transport choice and reducing disparities. Following Penchansky and Thomas' framework[26], as extended by Saurman[27], comprehensive food access depends on multiple factors: availability, geographic accessibility, adequacy or accommodation (e.g., store hours), affordability, acceptability (whether stores meet cultural preferences), and awareness. Our analysis focuses on geographic accessibility while acknowledging these broader dimensions.

To evaluate these spatial inequities, an approach that can be used to measure equality of access and equity between different groups was introduced in 2020: the Kolm–Pollak equally distributed equivalent (EDE), a measure similar to the Atkinson Index that is suitable for urban contexts[28–30]. As illustrated in Fig. 1, the EDE quantifies average access while penalizing greater distances, thereby prioritizing the needs of those with the poorest access. Unlike metrics that consider only distributional spread, such as the Gini index, which would prefer a solution where all residents are similarly far from an amenity over one where everyone is closer but with more variety in travel distances, the EDE incorporates both the average distance and its distribution. For example, if most residents live within a 10-min walk of a store but some must walk 30 min, the EDE would exceed the simple average to reflect this inequality. This makes it particularly suited for evaluating urban access, as it helps planners identify solutions that improve overall accessibility while reducing inequalities.

**Table 1 | Summary results for the 20 largest US cities, by 2020 urban population[65], sorted by EDE (best first)**

| City, state | EDE distance (km) | Rank (among 500 cities) | Urban population |
|---|---|---|---|
| New York, NY | 0.8 | 3 | 8,784,592 |
| San Francisco, CA | 1.0 | 7 | 871,136 |
| Miami, FL | 1.0 | 12 | 441,228 |
| Philadelphia, PA | 1.1 | 15 | 1,593,147 |
| Washington, DC | 1.1 | 17 | 684,900 |
| Chicago, IL | 1.1 | 20 | 2,733,239 |
| Seattle, WA | 1.1 | 26 | 726,482 |
| Los Angeles, CA | 1.3 | 56 | 3,849,235 |
| San Jose, CA | 1.6 | 99 | 993,779 |
| Denver, CO | 1.6 | 110 | 705,515 |
| San Diego, CA | 1.7 | 143 | 1,347,374 |
| Houston, TX | 1.9 | 209 | 2,215,641 |
| Charlotte, NC | 2.0 | 232 | 804,437 |
| Columbus, OH | 2.1 | 260 | 868,417 |
| Dallas, TX | 2.1 | 275 | 1,269,024 |
| Phoenix, AZ | 2.3 | 321 | 1,553,053 |
| Indianapolis, IN | 2.5 | 370 | 788,869 |
| San Antonio, TX | 2.6 | 388 | 1,381,080 |
| Fort Worth, TX | 3.3 | 449 | 865,707 |
| Jacksonville, FL | 4.1 | 483 | 834,225 |
| Austin, TX | 4.8 | 489 | 893,947 |

Miami (population: 441,228) is included due to its use as an example case throughout the paper, though it is not among the 20 largest cities. "EDE Distance (km)" indicates the equity-penalized mean distance (EDE) of residents to a grocery store. "Rank" indicates the rank of the city with respect to supermarket access among the 500 largest cities in the US (1 is best). See the Supplementary Materials for the full ranked list of 500 US cities.

Using the EDE, we evaluated access to supermarkets in the 500 largest cities in the United States. We use supermarket locations from the USDA's Supplemental Nutrition Assistance Program (SNAP) retailer database, defining supermarkets as full-service stores that typically offer a broad range of groceries, including fresh produce, meats, and deli items[10]. Table 1 shows the largest 20 cities in the study, along with their residents' inequality-penalized average access and their ranking among all 500 cities (the full dataset of the 500 cities is shown in the Supplementary Table). While food environments are complex and include many other important sources such as farmers markets, small grocers, and specialty stores, we focus on supermarkets due to data availability and consistency across all 500 cities in our study. When applied at a local scale, our method can optimize locations for other facility types, or multiple facility types simultaneously, given the critical role of smaller food outlets for improving food access in minority neighborhoods[11].

Having quantified these disparities in access, we now address the practical challenge of improving food environments in existing urban areas. We focus on two specific questions for each of the 500 largest cities in the US:

(1) If a city can encourage the opening of $k$ additional supermarkets, where should they be located to best improve equitable access?
(2) If we want to reach some level of equitable access (e.g., 15 min), how many additional supermarkets are required, and where should they be located?

The answers to these questions provide crucial insights about the scale of change required to retrofit car-oriented cities for both improved equity and sustainability. While cities cannot directly open supermarkets, research has shown that cities with successful grocery initiatives are those that have shown political leadership, engaged with public agencies,

collaborated across jurisdictions, and partnered with community-based nonprofits and existing small-scale food retailers[11,31,32]. This analysis, therefore, can inform multiple stakeholders: local governments using zoning and incentives, state and federal agencies developing funding programs like the Healthy Food Financing Initiative (HFFI), retailers identifying promising locations, and community organizations advocating for improved food access. Our optimization approach provides a rigorous, equity-focused method to guide these decisions across different urban contexts.

## Results

### Where should new facilities be located?

The first question we address is: if a city can open $k$ additional amenities, where should they be located to best improve equitable access? To answer this question, we developed an approach that minimizes the EDE: a metric that captures the average of a quantity (such as minimizing the distance to the nearest supermarket) but penalizes for inequality[33]. In this study, we applied this method to the 500 largest cities in the US.

For example, Fig. 2A shows a map of Miami, Florida, with shading to indicate the distance of residents to supermarkets based on the locations of supermarkets at the date of this study. In Fig. 2B, we show the recommended locations for five new supermarkets based on the traditional (mean distance minimizing) and our proposed equitable optimization approaches. Figure 2B is shaded according to the updated distance to the nearest supermarket given the optimal locations proposed by our method. Even though some of the added stores are sited at or near the same location under both approaches, our proposed placement leads to notable improvements for equality vs the traditional approach. In particular, there are often competing decisions of whether to choose a location that improves access by a few meters for many residents who already have good access, or to choose a location that dramatically improves access for fewer residents who currently have very poor access. When both of these solutions result in the same improvement in mean distance, our model chooses the latter, selecting the solution that improves the situation of those who are currently more disadvantaged.

In order to visualize the distributional effect of each intervention, we plot the access for each Block before and after the optimally-located supermarkets in Fig. 2C. The main graphic includes two marks for each Census Block in Miami: a red "×" corresponding to the mean-minimizing approach and a blue "○" corresponding to the inequality-optimizing approach. When a point is shown on the 1:1 line, it means that the Block's access has not changed as a result of the intervention. The further below the 1:1 line a point is, the more the corresponding Block's access has improved. No Block's access has gotten worse because we are only adding supermarket locations, not taking them away. This figure shows the distribution effect because it shows which Blocks experience the greatest improvement in access under each intervention. By comparing the traditional vs inequality-minimizing approaches, we see that the approach that seeks to optimize inequality tends to improve access to currently access-poor areas in comparison to the traditional approach. This is because the mean/average can be minimized by improving the access (reducing the distance) to any area.

The box plot in the upper left corner of Fig. 2C provides a visual representation of access statistics before and after each intervention. The inequality-minimizing (EDE-optimizing) method achieves nearly the same average and median access as the traditional (mean-minimizing) method, while more successfully targeting those with the poorest access in the baseline distribution. This effect is further analyzed and verified in the sister methods article[33].

### How many facilities are required?

Initially, we sought to answer the question of where a city should build additional amenities to improve equitable access. But this raises the question of how many are required to provide a certain level of equitable access? For instance, if a city is aiming for 10-min walkable neighborhoods, enabling people to reach the identified amenities within a 10-min walk of their residence, how many amenities are required, and where should they be built?

In the case of Miami, Florida, the city's supermarket access EDE is 1040 m, which is roughly a 12–13 min walk. However, for Miami to decrease this to 10 min (an EDE of 800 m), they would need an additional 12 stores, the locations of which are shown in Fig. 3. If they want to decrease the travel time to 5 min (400 m), they would need more than 100 additional supermarkets.

However, Miami was ranked 12th best out of the 500 US cities we studied. We proceeded to determine the number of additional supermarkets required for each of the 500 cities so that the EDE of the distance to the nearest supermarket was less than or equal to:

- the average EDE from all 500 US cities (2.29 km),
- 15 min (1200 m),
- 10 min (800 m),
- 5 min (400 m).

For the largest 20 cities in the US, the number of required supermarkets are shown in Table 2.

The number of supermarkets required for all 500 US cities are summarized in Fig. 4.

Of the 179 cities that are currently below the average level of walkable access, 73 cities require only one additional store to be on par with the average access for all 500 cities. Another 45 require two additional stores, and 43 more require between three and five new supermarkets. Jacksonville, FL and Miramar, FL are tied for the most stores required at 16.

To achieve an EDE value that represents a 15-min walk, the cities in the study require, on average, 18 additional supermarkets (at least 8640 supermarkets across the 487 cities where this is possible). Some cities are not far from this target. Twenty-three cities require only one supermarket, and 62 require between two and five additional supermarkets. 106 cities require between six and 10 additional supermarkets, while 132 require 11 to 20 more. With each 5-min improvement in access, the number of new stores required increases substantially.

The missing values in Table 2 indicate that the optimization model was "infeasible" for that city/access-target pair. This means that a solution could not be found based on the potential sites. This is a limitation arising from our use of Census Block Group centroids as the potential locations for new supermarkets. In some neighborhoods, the areas of the Block Groups were too large to provide sufficient options to achieve the goal. Developing a feasible model in these cases would require including more potential sites for a more uniform coverage of the city.

## Discussion

We sought to answer two specific questions for cities seeking to improve walkable access: where should new supermarkets be located to best improve equitable access, and how many are needed to achieve different access targets? Our analysis demonstrates that the answers vary considerably across cities (Fig. 4). For instance, 25% of the cities we studied could achieve 15-min access by adding five or fewer stores, while more than 90% of cities require more than 100 additional stores to achieve a 5-min city target. However, urgency is needed as these numbers will increase with further car-oriented urban development. The increase in required stores is not linear relative to the change in access target—on average, achieving 5-min access requires more than 10 times as many stores as improving from 15- to 10-min access.

For the question of where to locate new stores, our optimization approach demonstrates how cities can make this transition more

**Where should additional supermarkets be located and what is the impact on access and access equality?**
An example in Miami, Florida with five new supermarkets.

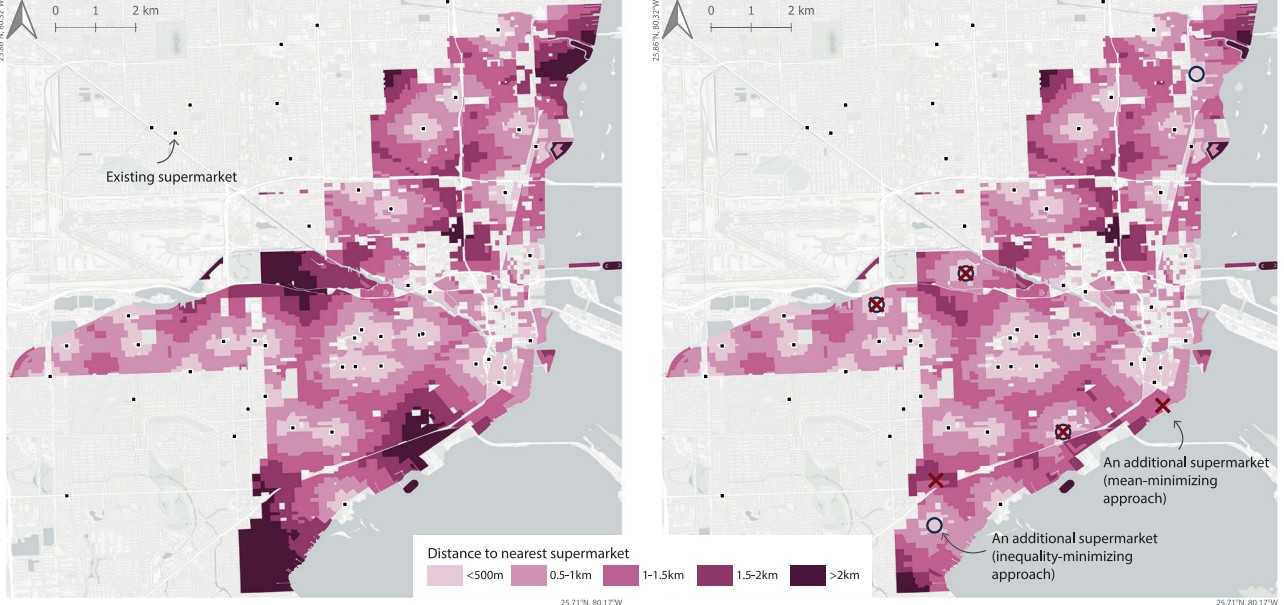

**A.** Access to supermarkets in Miami based on existing supermarkets.

**B.** The locations of five additional supermarkets and the updated (inequality-minimized) access in Miami.

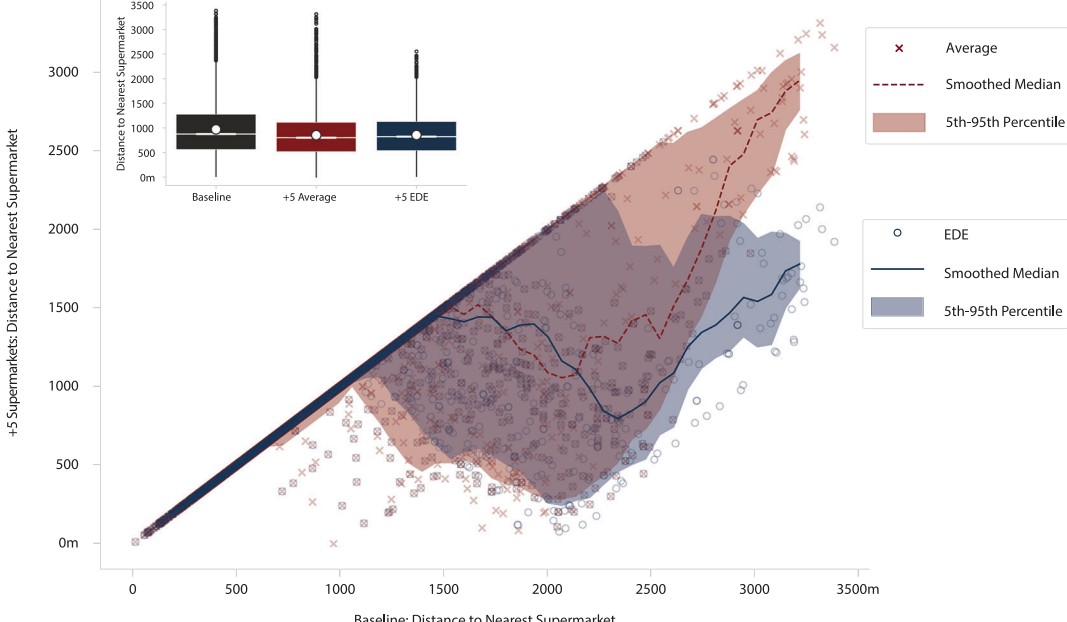

**C.** How are the benefits of the intervention approaches (two optimization models) distributed across the residents?

**Fig. 2 | Where should additional supermarkets be located to improve access and access equality?** Maps (**A**, **B**) show the access to supermarkets in Miami before and after the addition of five equality-optimizing supermarkets. Map **B** also shows the mean-minimizing locations of five additional supermarkets. The graphics in (**C**) show that the EDE-minimizing approach best targets the residents who currently have the worst access. The main graphic in (**C**) is a scatter plot where each point represents a Census Block (of the 3774 in Miami, FL), showing its access distance before (*x*-axis) and after (*y*-axis) each intervention−red crosses for the mean-minimizing approach and blue circles for the inequality-optimizing approach. The

box plot shows population-weighted means (indicated by a white circle with a black outline) and quartiles of before and after each intervention. In the box plot, the white center line represents the median, the box bounds represent the first (Q1) and third (Q3) quartiles, and the notches indicate the 95% confidence interval around the median. The whiskers extend to points within 1.5 times the interquartile range from the box edges, with points beyond this range shown as dots representing outliers. Basemap: © OpenStreetMap contributors, © CARTO, licensed under a CC BY Attribution 4.0 International License.

efficient by strategically locating new stores and is an advance from traditional inequality measures. While previous approaches using metrics like the Gini index focus only on reducing inequality, potentially at the cost of worse average access, our EDE-based optimization improves both the average distance and its distribution. For Miami, we

present optimal locations for both a modest intervention of five stores (Fig. 2B) and a more ambitious plan of twelve stores (Fig. 3) to achieve 10-min access.

This approach demonstrates how strategic location choices can better target areas with the poorest access while achieving similar

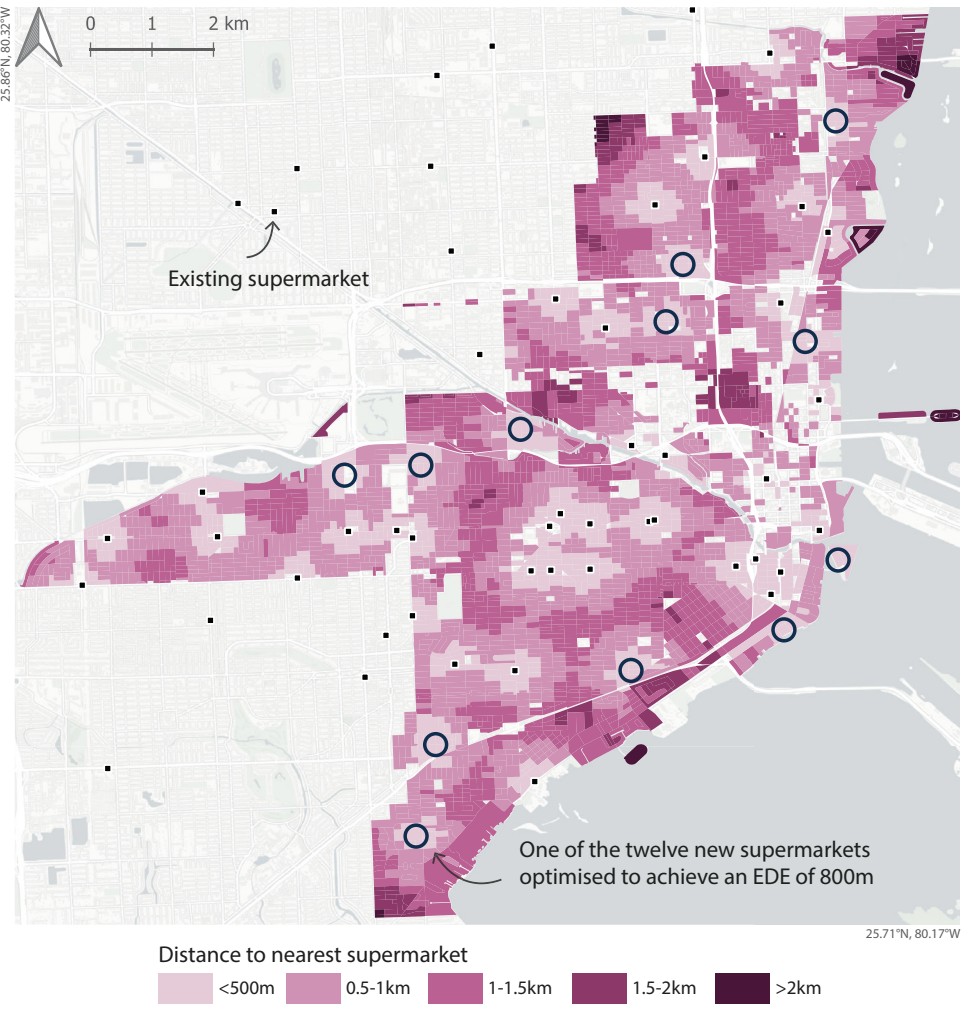

**Distance to nearest supermarket**

| | <500m | 0.5-1km | 1-1.5km | 1.5-2km | >2km |

**Fig. 3 | Equitable 10-min plan for Miami.** This map shows the 12 additional supermarkets that would be needed in Miami to achieve a supermarket access EDE (equally distributed equivalent) of 800 m, equivalent to a 10-min walk. The locations were generated using our inequality-optimizing model, which prioritizes improvements for residents with the poorest current access. The shaded background represents the updated access to the nearest supermarket after these additional stores are added. Basemap: © OpenStreetMap contributors, © CARTO, licensed under a CC BY Attribution 4.0 International License.

improvements in average access (Fig. 2C). Unlike methods that would accept uniformly poor access to achieve equality, our approach identifies solutions that improve overall accessibility while reducing extreme inequalities. This finding suggests that cities can make their retrofit efforts more efficient by explicitly considering these distributional impacts when planning new amenity locations.

While the x-minute city concept has gained popularity in urban planning, our method acknowledges that acceptable walking distances vary substantially across populations and contexts. Research shows this variability clearly—in the Netherlands, only 50% of respondents considered 400 m an acceptable distance to walk with groceries[15], while a New Zealand study found that willingness to travel to supermarkets varies by time of day, gender, age, and ability[34]. Rather than using binary access thresholds that create artificial cutoffs (where residents at 15.1 min are considered to lack access while those at 14.9 min have it), our approach focuses on continuously improving accessibility for all residents. By evaluating multiple distance thresholds, we demonstrate how the scale of intervention required varies with different accessibility targets. However, as we have argued previously[7], the ultimate goal should be improving accessibility rather than achieving arbitrary thresholds.

These findings provide an initial estimate of the challenge cities face in enabling active transport and transport choice. While the actual number of stores needed might vary depending on the availability of public transit and cycling infrastructure, the scale of change required underscores how car-dependent urban design limits transport options for many residents. Furthermore, the infrastructure costs of serving dispersed development are often subsidized by the broader community, making the case for more compact, walkable development patterns.

The practical application of these findings can inform multiple stakeholders working to improve food access across cities. Research underscores that successful initiatives require sustained, community-driven investment rather than relying on short-term interventions, which often fail to address structural inequities in food environments[19,35]. While cities can foster supportive environments through comprehensive planning frameworks and zoning reforms, achieving long-term success requires addressing persistent challenges such as the perceived risks and complexities of urban development[31], the need for regional and cross-jurisdictional collaboration to bridge uneven capacities across municipalities[32,36], and the importance of strengthening local food infrastructure through targeted policies and investment[37]. Focusing exclusively on attracting large supermarkets risks overlooking opportunities to enhance food access through existing small-scale retailers, which often serve minority and relatively low-income communities[11,35]. Evidence from Detroit FRESH and other initiatives highlights that meaningful community engagement,

**Table 2 | Summary results for the 20 largest US cities sorted by rank**

| City, state | Rank | Additional supermarkets | | | |
| --- | --- | --- | --- | --- | --- |
| | | Average (2290 m) | 1200 m | 800 m | 400 m |
| New York, NY | 3 | 0 | 0 | 2 | 440 |
| San Francisco, CA | 7 | 0 | 0 | 9 | 127 |
| Philadelphia, PA | 15 | 0 | 0 | 27 | 329 |
| Washington, DC | 17 | 0 | 0 | 15 | 182 |
| Chicago, IL | 20 | 0 | 0 | 43 | 553 |
| Seattle, WA | 26 | 0 | 0 | 27 | 311 |
| Los Angeles, CA | 56 | 0 | 8 | 186 | 2216 |
| San Jose, CA | 99 | 0 | 19 | 121 | – |
| Denver, CO | 110 | 0 | 14 | 72 | – |
| San Diego, CA | 143 | 0 | 51 | 201 | – |
| Houston, TX | 209 | 0 | 112 | 469 | – |
| Charlotte, NC | 232 | 0 | 126 | – | – |
| Columbus, OH | 260 | 0 | 81 | 270 | – |
| Dallas, TX | 275 | 0 | 63 | 256 | – |
| Phoenix, AZ | 321 | 0 | 80 | 358 | – |
| Indianapolis, IN | 370 | 3 | 121 | 462 | – |
| San Antonio, TX | 388 | 7 | 181 | 884 | – |
| Fort Worth, TX | 449 | 8 | 103 | 371 | – |
| Jacksonville, FL | 483 | 16 | 362 | – | – |
| Austin, TX | 489 | 5 | 102 | 445 | – |

The last four columns indicate the number of additional supermarkets needed in each city to achieve a given level of access. "Average" indicates the average level of access across all 500 cities in the study (2290 m), so Indianapolis, ranked at 370, is the first city in the list to require any additional supermarkets to reach the average level of access. The missing values indicate that the potential sites (Census Block Group centroids) provided to the model were insufficient for reaching the target level of access.

combined with long-term financial and policy support, is critical for the success of these efforts[19,35].

Our optimization approach can contribute to these multi-faceted efforts by identifying priority areas where new food retail outlets would have the greatest impact on improving access. Our results should be interpreted as identifying areas of need and opportunity, rather than as direct recommendations for store placement. Local governments could use this model to guide targeted incentive programs, while state and federal agencies might allocate funding—such as through the HFFI—to underserved areas[32,37]. Community organizations could advocate for supermarket placements that align with neighborhood needs, fostering collaboration between public and private sectors[31,36]. Importantly, achieving lasting improvements requires moving beyond disconnected solutions, such as the addition of isolated stores, to comprehensive strategies that combine optimizing new supermarket locations with investments in existing local food retailers, developing robust regional food infrastructure, and ensuring community leadership in decision-making processes[19,32,36].

Although our analysis focuses on geographic accessibility, it represents just one dimension of comprehensive food access[26,27]. The other dimensions are crucial: availability (whether stores have sufficient capacity and appropriate products), accommodation (such as store operating hours), affordability (both of food and transportation costs), acceptability (whether stores meet cultural and social preferences), and awareness (whether residents know about available food options). Even within geographic accessibility, temporal patterns substantially shape access—store operating hours, variation in personal mobility throughout the day, and cyclical patterns of food purchasing tied to benefit distribution all affect when people can reach stores[38]. Research shows that people's food shopping patterns are influenced by multiple factors beyond simple proximity—including store quality, price, safety, and social networks[39]. Residents often travel outside their immediate neighborhoods and visit multiple stores to balance these considerations with their cultural preferences and economic constraints[39].

These complexities reflect a deep interdependence between urban design, sustainability, retail trends, and social equity. The consolidation of food retail into larger formats[40] both responds to and reinforces car-dependent urban form—as stores become larger and fewer, walking becomes less viable, which further increases car dependency. This pattern particularly affects communities that already face multiple barriers to food access. Our findings about the number of stores needed to achieve walkable access underscore the scale of this challenge—the substantial investment required in many cities reflects decades of car-oriented development and retail consolidation. Breaking this cycle requires coordinated action across scales, from neighborhood-level support for smaller food retailers to regional policies that address broader patterns of retail distribution. While improving walkable access alone cannot solve food inequities, it represents an important step toward both more sustainable and more equitable urban environments.

While this study focuses on food access, our findings and methods have broader implications for sustainable and equitable urban planning. The optimization approach we present could inform the placement of any amenity where equitable access matters—from healthcare facilities to green space to civic services. Moreover, our results highlight a fundamental tension in retrofitting car-oriented cities: the more ambitious our goals for walkable access, the more facilities are required to overcome decades of sprawling development. This challenge extends beyond food retail to all essential services that communities need for daily life. Cities working to reduce car dependency and improve equity must therefore consider both the spatial distribution of multiple amenity types and the mutually reinforcing relationship between urban form and service provision[41,42]. This work is particularly salient as cities worldwide seek to improve distributional justice[17,18,43] and rebuild community resilience following disruptions[44–46]. Success requires moving beyond single-issue solutions to understand how different aspects of urban accessibility—whether to food, transport, healthcare, or other services—work together to enable both sustainable and equitable communities[13].

## Methods
In this paper, we seek to evaluate and optimize access (and access inequality) to grocery stores across the 500 largest cities in the US. In this section, we describe our data[47], provide background information on the metric we use to quantify inequality-penalized access, and present our optimization models.

### Cities
The cities analyzed in this study are those included in the Centers for Disease Control and Prevention (CDC) 500 Cities Project (superseded by their PLACES project[48]), which focused on the largest 500 US cities. While the CDC project focused on chronic disease measures, our study repurposes this established set of cities to explore accessibility and equity in food retail distribution.

While our analysis focuses on cities defined by administrative boundaries, we recognize that urban areas often function as part of larger metropolitan systems with interconnected food environments that transcend municipal borders. This administrative boundary approach provides a standardized method for analyzing accessibility across multiple cities, but may not fully capture the functional food landscape that residents experience. Metropolitan-scale analysis would offer additional insights into regional food access patterns, particularly for residents near city edges who may regularly cross-jurisdictional boundaries for groceries or as part of their daily routine.

How many supermarkets are needed across the largest 500 US cities
to achieve different levels of access

Only 51 cities could achieve 400m access
based on the potential new supermarket locations

367 cities could achieve 800m

487 cities could achieve 1200m

47 cities require 3-5 more supermarket to reach an EDE of 1200m

73 cities require 1 additional supermarket to reach the average EDE

40 cities already have an EDE of 1200m

Access target: 400 m  800 m  1200 m  Average (2290 m)

|  | Average | 1200 meters | 800 meters | 400 meters |
|---|---|---|---|---|
| Min | 0.00 | 0.00 | 0.00 | 3.00 |
| Mean | 0.96 | 17.74 | 49.48 | 144.55 |
| Max | 16.00 | 362.00 | 884.00 | 2,216.00 |

Number of supermarkets required per city

**Fig. 4 | Number of additional supermarkets needed to achieve equitable access.** Nearly every city can reach the average level of access by adding fewer than 10 supermarkets. The outlook is less positive for more ambitious access levels. The statistics in the table represent cities for which the target access is feasible.

Although we partially address this by including stores within a 5-km buffer beyond city limits, a comprehensive metropolitan analysis could reveal different optimal store placement patterns, especially in highly fragmented urban regions where municipal boundaries may not align with functional economic areas. Future work examining food access at the metropolitan scale could provide complementary insights about how regional planning and cross-jurisdictional coordination might improve equitable access beyond what individual cities can achieve independently.

Following the methodology of ref. 7, we focus our analysis on Census Block Groups within these cities' administrative boundaries with densities greater than 200 housing units per km². This density threshold is equivalent to moderate density in a suburban area and captures areas where walkable access is most feasible and infrastructure investments are most efficient. The analysis thus avoids recommending stores in locations where they would likely be economically unsustainable. In the US, the smallest geographical reporting unit in the census is the Census Block. In urban areas, Census Blocks often correspond to city blocks, bounded on all sides by streets. The second smallest geographic unit is the Census Block Group, which typically contains 600–3000 residents.

Although our analysis is constrained to the administrative boundaries of these cities, we account for aspects of the functional nature of urban systems by incorporating grocery store locations within a 5-km buffer zone beyond city limits. This approach acknowledges that urban areas extend beyond their formal boundaries, and amenities located just outside these limits may serve city residents.

**Measuring access**

We measure access as the distance to the nearest amenity. This means that our approach is not considering the demand for a particular amenity or its capacity to serve that demand, and is a limitation that we will seek to address in the future. However, while not the only requirement, proximity to services is necessary for access[26,27].

**Network distances.** To complete our analysis, we required walking distances from the centroid of each US Census Block (the smallest census unit for the US) to all existing grocery stores and potential store locations. We calculated these distances using Open Source Routing Machine (OSRM)[49] via the method described by Logan et al.[50]. Using data from Open Street Map[51], OSRM calculates the shortest path along the road network from origin to destination. This method is more accurate than Euclidean or Manhattan distances because it is based on actual roadways, navigating around geographical barriers, such as freeways, waterways, and railroad tracks, as required. However, this method does not account for the suitability or quality of the walking environment (see ref. 52 for a discussion) but only whether a route exists.

**Grocery store locations.** We use existing supermarket locations from the USDA's Food and Nutrition Service 2021 SNAP database[53], which catalogs retailers authorized to accept SNAP benefits. Although the 2021 data is no longer available, more recent data are available at https://usda-fns.hub.arcgis.com. Following Kolak et al.[10], we define supermarkets as full-service stores that typically offer a broad range of groceries, including fresh produce, meats, and deli items. The store should have five or more checkout lanes[10]. The criteria were chosen to include stores offering a variety of healthy food options[54]. We identified supermarkets using a list of national and regional chains that we compiled, explicitly excluding convenience stores, gas stations, and smaller retail formats. While this approach may exclude some smaller stores that sell fresh

produce, it provides a consistent classification method that can be applied at scale across the 500 cities. Our focus on SNAP-authorized retailers ensures we capture stores that are accessible to residents who depend on these benefits.

While food environments include a range of sources such as farmers markets, small grocers, and specialty stores, we analyze supermarkets due to consistent data availability across the 500 cities. The SNAP retailer database provides standardized data nationwide, and supermarket chains can be reliably identified within this database. Our method could be adapted to optimize locations for other facility types or to consider multiple facility types simultaneously, provided consistent location data is available.

This flexibility means the model could optimize locations for multiple facility types simultaneously—for example, considering both supermarkets and smaller grocers together, with different weights or service areas for each type. Such an approach could help planners balance the complementary roles of different retail formats in creating comprehensive food environments.

We include supermarkets within a 5-km radius of the city boundary to ensure we capture stores that residents near city boundaries might reasonably access, even if those stores lie in neighboring jurisdictions. This approach acknowledges that urban areas extend beyond their formal boundaries, and amenities located just outside these limits may serve city residents. This is consistent with the analysis of Logan et al.[7].

**Potential grocery store locations.** We used the centroids of US Census Block Groups from the 2020 US Census to geographically cover each city with potential store locations. After Blocks, Block Groups are the second most granular geographic unit captured in the US Census.

**Population data.** The population of each Census Block was based on the 2020 US Census and exported from the IPUMS National Historical Geographic Information System[55].

### Inequality metric
The Kolm–Pollak EDE provides a way to measure both average access and inequality in its distribution. Unlike metrics that only capture spread (like the Gini index), the EDE asks: "What uniform level of access would make a community indifferent between that level and their actual unequal distribution?"[28]. This makes it particularly suitable for urban accessibility, as it incorporates both the average distance and a penalty for areas with poor access. For example, if most residents live close to amenities but some face much longer distances, the EDE will exceed the mean to reflect this inequality.

The EDE was introduced as the only metric satisfying key properties identified by the environmental justice community for comparing distributions of environmental burdens[28]. A recent study demonstrates its application to urban amenity access, showing how different levels of inequality aversion affect city rankings[29].

Like other EDEs, the Kolm–Pollak EDE depends on a user-defined parameter, $\epsilon \in \mathbb{R}$. For undesirable quantities like distance or pollution, $\epsilon < 0$, and the EDE is always at or above the mean, with larger $|\epsilon|$ values representing greater aversion to inequality. We used the EDE distance (in meters) with $\epsilon = -1$ to quantify the level of access of a community to grocery stores. For a given city, let $R$ represent the set of Census Blocks and let $p_r$ represent the population of Block $r \in R$. Let $z_r$ represent the walking distance (in meters) of Block $r \in R$ to the closest grocery store. The Kolm–Pollak EDE distance of the residents of the city to supermarkets is,

$$\mathcal{K}(\mathbf{z}) = -\frac{1}{\kappa}\ln\left[\frac{1}{T}\sum_{r \in R} p_r e^{-\kappa z_r}\right], \tag{1}$$

where $\mathbf{z}$ is the vector of distances, $T := \sum_{r \in R} p_r$ is the total population, and $\kappa := \alpha\epsilon$, where

$$\alpha = \frac{\sum_{r \in R} p_r z_r}{\sum_{r \in R} p_r z_r^2}. \tag{2}$$

The aversion to inequality, $\epsilon$, is scaled to the problem data via $\alpha$, so $\kappa$ is the appropriately scaled aversion to inequality.

The EDE inherently weights access by population—locations serving more people will tend to improve the metric more. However, the inequality penalty ensures we don't overlook smaller populations with poor access, creating a balance between serving dense areas and maintaining equity.

### Optimization
Many models aimed at incorporating equity in facility location optimization have been proposed[56,56–60]. However, the computational expense required to solve these models has limited their practical value. Modern optimization solvers can handle either nonlinear functions or integer-valued variables in large problems, but not both simultaneously. Since facility location models require integer variables (0/1-valued variables that indicate whether to place a facility at a potential location), including a nonlinear equity metric severely restricts the size of problems that can be solved[61].

We overcame this limitation by developing a linear model to optimize the nonlinear EDE. Our approach scales to large, city-sized instances, enabling us to analyze optimal facility locations across all 500 cities in our study. While the technical details are presented in a sister article[33], here we demonstrate how this method can inform practical planning decisions about where to locate new facilities to improve equitable access.

Adding to the notation defined above, let $S$ represent the set of existing and potential supermarket locations, and let $C \subseteq S$ represent the set of existing (current) supermarket locations. Let $d_{r,s}$ represent the walking distance (in meters) between Block $r \in R$ and location $s \in S$. Our decision variables are all binary:

$x_s := 1$ if a supermarket is placed at location $s \in S$, 0 otherwise;
$y_{r,s} := 1$ if Block $r \in R$ is assigned to service location $s \in S$, 0 otherwise.

As a function of the vector, $\mathbf{y}$, of $y_{r,s}$ variables, the EDE is,

$$\mathcal{K}(\mathbf{y}) = -\frac{1}{\kappa}\ln\left[\frac{1}{T}\sum_{r \in R} p_r e^{-\kappa\sum_{s \in S} y_{r,s} d_{r,s}}\right]. \tag{3}$$

We can equivalently minimize the so-called linear proxy[33],

$$\overline{\mathcal{K}}(\mathbf{y}) = \sum_{r \in R}\sum_{s \in S} p_r y_{r,s} e^{-\kappa d_{r,s}}, \tag{4}$$

and then convert the optimal objective value to an EDE score:

$$\mathcal{K}(\mathbf{y}) = -\frac{1}{\kappa}\ln\left(\frac{1}{T}\overline{\mathcal{K}}(\mathbf{y})\right).$$

Note that $\kappa$ depends on the distribution of distances, making $\kappa$ a nonlinear function of the decision variables. In order to obtain an optimization model that can be solved for large instances, we must treat $\kappa$ as a constant. We used the current distributional access (with existing store locations) to approximate the value of $\alpha$ that corresponds to the optimal distribution of distances. Our computational tests indicate that treating $\kappa$ as a constant has very little impact on optimal solutions. For a detailed discussion of our strategy for approximating $\alpha$, please see our companion methods article[33].

**Question 1.** To answer the question of how to optimally locate $k$ additional supermarkets, we minimize the EDE linear proxy in the objective function, (5):

$$\text{minimize } \overline{\mathcal{K}}(\mathbf{y}) = \sum_{r \in R} \sum_{s \in S} p_r y_{r,s} e^{-\kappa d_{r,s}}, \tag{5}$$

$$\text{subject to } \sum_{s \in S \setminus C} x_s = k; \tag{6}$$

$$\sum_{s \in S} y_{r,s} = 1, \quad \forall r \in R; \tag{7}$$

$$y_{r,s} \leq x_s, \quad \forall r \in R, s \in S; \tag{8}$$

$$x_s = 1, \quad \forall s \in C; \tag{9}$$

$$x_s, y_{r,s} \in \{0,1\}, \quad \forall r \in R, s \in S. \tag{10}$$

As noted above, we can convert the optimal objective value to an EDE distance to determine the optimal level of access that can be achieved by adding $k$ stores. Constraint (6) ensures the correct number of new supermarkets are opened. Constraint (7) ensures that every Census Block is assigned to a single store, while (8) ensures that the assigned store location is open. Constraint (9) keeps all existing stores open, and (10) enforces the binary requirement on the indicator variables.

**Question 2.** In the model that answers how many (and where) stores should be opened to achieve a given level of equitable access, the EDE linear proxy is included as a constraint. Suppose we are aiming for a level of equitable access of no more than $\ell$ meters. We must convert $\ell$ to the same units as the linear proxy to serve as the upper bound on the access constraint: $L = Te^{-\kappa\ell}$. The model that answers our second question is:

$$\text{minimize } \sum_{s \in S \setminus C} x_s, \tag{11}$$

$$\text{subject to } \sum_{r \in R} \sum_{s \in S} p_r y_{r,s} e^{-\kappa d_{r,s}} \leq L; \tag{12}$$

$$\sum_{s \in S} y_{r,s} = 1, \quad \forall r \in R; \tag{7}$$

$$y_{r,s} \leq x_s, \quad \forall r \in R, s \in S; \tag{8}$$

$$x_s = 1, \quad \forall s \in C; \tag{9}$$

$$x_s, y_{r,s} \in \{0,1\}, \quad \forall r \in R, s \in S. \tag{10}$$

The objective function, (11), minimizes the number of new stores, while (12) ensures the desired level of access is achieved. The rest of the constraints are the same as in the previous model.

### Computing environment
We implemented the models in Python using the optimization modeling language Pyomo[62,63], and solved the models using the linear mixed-integer optimization solver, Gurobi[64]. We solved most instances on a high-performance computing cluster, an Advanced Micro Devices (AMD) 7502 CPU processor with 64 cores and 512 GB of memory, allocating one out of the 64 available cores to each instance. The New York instances required more memory. For those, we used an AMD 7502 CPU processor with 64 cores and 2 TB of memory.

### Limitations
Our results may either underestimate or overestimate the number of stores required. The use of an inequality-penalized average means some areas will have to travel further than the target distance; a stricter definition requiring no resident to exceed x minutes would need substantially more stores. Additionally, our analysis represents a snapshot in time and does not account for ongoing urban development and sprawl. Conversely, by focusing solely on walking access, we may overestimate required stores in cities where cycling infrastructure or public transport effectively extends catchment areas. While our optimization methodology could incorporate these alternative transport modes if travel time data were available, this study provides an initial estimate based on walking access.

Several methodological choices were made to enable analysis at this scale, and these introduce important limitations. Following Penchansky and Thomas' framework[26,27], our analysis addresses only one of six dimensions of access: geographic accessibility. We do not capture availability (sufficient capacity and appropriate products), accommodation (store hours), affordability (food and transportation costs), acceptability (cultural and social preferences), or awareness (knowledge of options). Even within geographic accessibility, our measure is simplified—we consider only distance to the nearest store and do not account for factors like sidewalk quality, street lighting, safety, or topography that affect walking viability. By using Census Block Group centroids as potential store locations, we may miss viable sites, particularly in areas with large Block Groups. While this simplification was necessary for computational tractability across 500 cities, it is not inherent to the method.

Our focus on SNAP-authorized supermarkets provides only a partial view of food environments. While these retailers typically offer broad food options[10], many communities are also served by farmers markets, specialty stores, and smaller grocers that our analysis excludes. We do not capture temporal variations in accessibility—recent research shows that comfort with walking varies substantially by time of day, gender, and ability[34]. Our method could incorporate these additional retailers and temporal patterns if consistent data were available across all cities.

While we use a density threshold of 200 housing units per $km^2$ to focus on potentially viable locations, we do not consider economic factors like purchasing power, competition, or operating costs. Our method inherently weights access by population through the EDE calculation, favoring denser areas while ensuring equity through the inequality penalty. However, successful implementation would require considering market forces, community preferences, and political constraints beyond our analysis scope.

This analysis could be extended in several directions: to a metropolitan scale, incorporating multiple facility types, and considering temporal variations in accessibility. Future work could also integrate other dimensions of access, analyze distributional impacts between socio-demographic groups[29], and incorporate additional measures of urban form and economic viability.

### Reporting summary
Further information on research design is available in the Nature Portfolio Reporting Summary linked to this article.

## Data availability
The datasets generated and analyzed during this study are publicly available via Zenodo at https://doi.org/10.5281/zenodo.15261330[47].

## Code availability

The code used for data processing, equitable facility location optimization, and visualization is publicly available on GitHub at https://github.com/drewhort/equitable_facility_location, and archived via Zenodo at https://doi.org/10.5281/zenodo.15264182.

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

## Acknowledgements

We used the Positron [no labels] basemap style provided by CARTO under the Creative Commons Attribution 4.0 license. Basemap data derived from OpenStreetMap and OpenMapTiles. This work used computing resources at the Center for Computational Mathematics, University of Colorado Denver, including the Alderaan cluster, supported by the National Science Foundation award OAC-2019089 (D.H., E.S.).

## Author contributions

D.H.: methodology, software, validation, formal analysis, investigation, writing—original draft. T.L.: conceptualization, data curation, formal analysis, investigation, resources, writing—original draft, writing—review and editing, visualization. E.S.: methodology, formal analysis, resources, writing—original draft, supervision. D.S.: methodology, writing—original draft, writing—review and editing.

## Competing interests

T.L. has a financial interest in the software firm Urban Intelligence Ltd. The remaining authors declare no competing interests.
