## [Transparent Peer Review file · Nature Communications]

Hundreds of supermarkets needed across the United States to achieve walkable cities

Corresponding Author: Dr Tom Logan

Version 0:

Reviewer comments:

Reviewer #1

(Remarks to the Author)

This work brings a very interesting approach to assessing the unequal distribution of opportunities within the urban space. The work builds on the Kolm-Pollak equally distributed equivalent (EDE) approach based on the Atkinson index. This is undoubtedly a novel approach to assessing spatial issues in urban systems. We believe this approach is promising for mapping spatial inequalities in cities (access to opportunities and exposure to hazards). However, it has conceptual weaknesses that can be overcome. Our purpose with this review is to help the authors find ways to overcome this challenge, or at least express the limitation of the analysis with more clarity and set the conceptual basis for advancing their ingenious method.

Conceptual challenges

The omission of urban density

The Atkinson index represents social welfare as an average utility. This is particularly relevant to the distribution of resources, such as income. The higher the mean income, the higher the social welfare; this is the idea behind it. Such an idea has been implemented through redistributive tax policies, and good evidence suggests it has been helpful in achieving more just and peaceful societies. However, the problem with cities and access to opportunities is different. It is fundamentally a spatial problem. The more centrally located one is in an urban system, the better the possibility of accessing more opportunities (see Space Syntax theory). The problem is that the lack of a central location cannot be easily compensated by subsidies, as is the case with income. Transport subsidies can alleviate the situation, but the disadvantage persists in the form of extra time (see Hägerstrand 1970 "What About People in Regional Science"). Indeed, in the last part of the abstract, the authors recognise that achieving the goal of a mean distribution of opportunities in contemporary cities is unfeasible. They claim it is because of a car-oriented urban design (which is part of the truth). However, premodern cities also had the same issue; people living in the periphery had a spatial disadvantage in accessing opportunities.

In contemporary cities, the massification of cars has expanded accessibility to large peri-urban areas, decreasing urban density. The article acknowledges this issue (lines 162-3), but the analysis does not capture this crucial aspect. Many people choose to live in low-density areas because they believe access to more spacious housing, far from noisy, more compact urban cores, is worth paying with poorer access to opportunities (more distance, time, and transport costs). City planners know this is problematic, but it has been challenging to tackle in market-oriented societies where freedom of choice is highly regarded. However, making cities more compact and, therefore, more efficient remains a central endeavour for city planners. Taxpayers already subsidise this lifestyle in some way by maintaining roads and other elongated infrastructure at the state's expense (see, for example, the numerous studies addressing the subject of "the cost of sprawl"). Is this fair? In this light, what does "distributional justice accessibility" mean (line 215)? Can this, for example, be explained using Rawls' idea of justice as fairness (to mention one widely accepted definition of justice)? The authors claim in lines 178-80 that the model provides optimal equitable facility locations, enabling the transition to not only be efficient, but just. It would be interesting if the authors described their idea of justice and efficiency (just for whom and efficient for whom).

Due to the low population density, many areas may not have grocery shops because the location is not economically feasible for the shop owner. Should the taxpayer then subsidise the location of non-economically feasible supermarkets in low-density areas because everybody should live within x minutes' distance from a grocery shop? It seems illogical to think this can be the author's idea, but if it is not the case, why was population density not part of the analysis?

The analysis has significant limitations without considering population density (or at least the number of served/non-served citizens). It presents a very coarse picture of underserved areas but does not provide an empirical base for the location of economically viable grocery stores. Such an analysis may require including different measures of urban density such as population, network density, floor space index, etc. Some areas with enough density to make grocery shops economically feasible may be, in fact, deprived of such service. If the authors can depict that, they are making a valuable contribution to planning authorities and helping grocery shop owners find economically feasible new locations. It can be challenging to include that for an analysis of 500 cities. Still, the limitation needs to be addressed more explicitly, explaining why density has not been possible to include in this current analysis. This inclusion can be an avenue to expand this research in the future. These limitations have been somehow addressed in section "Measuring Access" (lines 241-61), but they need to be more central in the argument. It is not a minor thing; density is fundamental for efficiency in urban areas.

There is a need to clarify the concept of city.

The sample selection seems to be based on the 500 most populous municipalities (cities, according to the paper). The challenge here is that cities are regional phenomena (see the concept of "functional urban area"), and exceptionally, access to opportunities is constrained by administrative boundaries. It seems the authors are aware of this phenomenon by including grocery store locations within a 5 km radius of the administrative boundaries of the cities in their analysis (lines 254-256). This is depicted in Figure 1, which illustrates that existing supermarkets outside the boundaries of Miami serve a part of Miami's population. However, we recommend that authors clarify the definition of a city and its relationship with the phenomenon of functional urban area in the 'Main' section of the manuscript since the performed analysis is eventually constrained to the administrative jurisdiction of the examined cities. This will serve the narrative's clarity and comprehension.

Minor comments

According to the proposed EDE optimisation model, the authors use Figure 3 and Table 2 to present the generated results regarding how many supermarkets are needed across the largest 500 US cities to achieve different access levels. While these figures provide a good overview for the reader, we recommend that alongside 'Average', it is also indicated that this category refers to the distance of '2290 m'. We believe that this will improve the comprehension of the figure and table.

Lines 165-172 elaborate on how public transportation usage can affect the generated results, but what about cycling? In specific cases, cycling can be an even more influential mobility choice with respect to accessibility to services than public transport. Therefore, we suggest articulating cycling as a potential mobility mode that can extend the studied amenities' catchment area.

Fabio Hernández Palacio and Todor Kesarovski

(Remarks on code availability)

Reviewer #2

(Remarks to the Author)

(Remarks on code availability)

NCOMMS-24-27165-T

Reviewer #3

(Remarks to the Author)

OVERVALL REFLECTION

The authors address a timely topic, use a deft methodological approach, and offer intriguing findings. That said, I do not entirely agree with the implications of the authors' findings, and I hope my reflections help the authors strengthen the paper (especially the discussion section). I recommend that the paper be published with minor revisions.

The authors investigate the degree to which cities can be retrofitted to achieve the so-called "15-minute city" ideal that has reappeared in popular discourse. Loosely speaking, a 15-minute city affords its residents the opportunity to reach life's amenities within a 15-minute walking/bicycling distance of their homes. The 15-minute city idea has been critiqued in the literature as lacking a theoretical framework and is extolled without sufficient empirical evidence for why and how such a vision might be implemented in cities. In this paper, the authors explore the feasibility of the 15-minute city idea by examining the residents' spatial access to a particular type of food retail in the 500 largest cities in the United States using the Kolm-Pollak equally-distributed equivalent (EDE) index. The use of the Kolm-Pollak equally-distributed equivalent (EDE) index is well executed, and illustrates a novel way of understanding inequitable access in a particular type of food retail across the US.

NOTEWORTHY RESULTS

The article, with the provocative title “Hundreds of supermarkets needed across the United States to achieve walkable cities,” reports that 25% of the studied (500) cities in the US could, in theory, provide 15-minute access to supermarkets by adding fewer than 5 new supermarkets. The authors report that to reach the 15-minute ideal, cities in the study require on “average 18 additional supermarkets. [...] Some cities are not far from this target. 23 cities require only one supermarket and 62 require between two and five additional supermarkets. 106 cities require between six and ten additional supermarkets, while 132 require 11 to 20 more.” Two look at two examples from the author tables, NYC needs no supermarkets to become a 15-minute city while Jacksonville, Florida needs to add the most supermarkets (362) to become a 15-minute city.

Notwithstanding the provocative title of the article, the actual data from the article suggest that a relatively small number of additional supermarkets are necessary for many US cities to reach the 15-minute city ideal. Six of the 20 largest cities in the US - NYC, San Francisco, Philadelphia, Washington, Chicago, and Seattle - have already reached this quantitative/spatial ideal. As the authors note, 25% of cities in their dataset could reach this ideal by adding 5 or fewer supermarkets. Still, the findings are sobering - because it is extremely difficult to convince a supermarket proprietor to move (or, stay) in a neighborhood that they do not wish to be in for a variety of reasons.

SIGNIFICANCE AND ORIGINALITY

This is a creative and original piece of work that adds empirical rigor to the idea of the 15-minute city. The methodological approach and findings are useful in illustrating that the 15-minute city ideal is already in place for some cities and out of reach for others (in so far as access to a particular type of food retail is concerned). The methodological approach can be adapted for each places gauge where food retail amenities (or, other amenities) might be added to create a more equitable food retailscape.

Recommendation: If you intend to speak to urban planners, consider reviewing/locating your work in urban planning literature that has documented the challenges in food retail (Raja et al, 2008), and how difficult it is to bring supermarkets into US cities (Kami Pothukuchi 2005). Consider the following journals: Economic Development Quarterly, Journal of the American Planning Association; Journal of Planning Education and Research; Journal of Agriculture, Food Systems, and Community Development; Environment and Planning.

THEORETICAL FRAMING

The authors reference the presence of “food deserts” in the United States to justify the significance of their work. This, in my judgment, undermines their own excellent focus (and analysis) on equitable access. The food deserts literature has been heavily criticized for its lack of theoretical framing, weak empirical analyses, and deficit-based view of Black and brown communities in the United States. In fact, the authors’ use of the Kolm-Pollak EDE, which focuses on equity (not equality), is a welcome departure from the food deserts literature.

Recommendation: For stronger theoretical framing, I recommend reviewing the extensive food apartheid literature and the literature on white accumulation which explains why amenities are located just outside of city boundaries (in inner-ring suburbs). Edward Goetz writes about this in the context of housing, for example. See this short piece by Food Equity scholars which highlights how food apartheid and accumulation work hand-in-glove in food retail disparities <https://civileats.com/2022/05/24/op-ed-east-buffalo-needs-community-driven-structural-investments-not-fly-in-fly-out-charity/>). Frankly, I would be delighted to see the analysis of this paper executed for metropolitan areas (rather than cities) because most food retail inequity is more pronounced across the city-suburb scale than within cities [The authors are already getting at this a bit because they have included supermarkets within 5 km boundary of cities in their analysis, but the theoretical reasoning for doing so is absent/weak.]

METHODS AND ANALYSIS

As noted earlier in this review, the use of the Kolm-Pollak equally-distributed equivalent (EDE) index is clever. Empirical work of the kind offered in this paper is much needed especially because (some) government officials fail to interrogate what reaching a 15-minute ideal might mean in a particular urban-regional context.

Recommendations for improvement

On page 5, the authors write that “we use the words grocery stores and supermarkets interchangeably; by both we mean stores that sell food, including fresh produce, and are larger than convenience stores or gas stations”. The authors also point to an earlier article (Logan et al, 2022) for their definitions/use of supermarket data. Additionally, the authors note (on page. 17), “we use existing supermarket locations within a 5km radius of the city from the USDA’s Food and Nutrient Service SNAP database available on ArcGIS Hub.” The available information does not fully explain the type of store that the authors have included in the analysis - and this matters a great deal for the interpretation/discussion of their paper. Presuming that the analysis is based on the USDA database that provides locations of all stores that are authorized to accept Supplemental Nutrition Assistance Program (SNAP) benefits, I would recommend the following clarifications:

1. Clarify that the paper is based only on retail stores that accept SNAP benefits (not all supermarkets/grocery stores; there are some supermarkets/grocery stores that may not accept SNAP benefits). In my judgment this is not a problem - in fact, if there are stores that do not accept SNAP they are by definition inaccessible to people who depend on SNAP benefits. The authors simply need to say this.
2. Additionally, I would recommend that the authors report precisely the type of SNAP food retail that they have included in

their analysis. As I understand it, the USDA SNAP retail location database includes multiple categories: superstores (e.g. Walmart, an increasingly common source of fresh produce for low-income people), supermarkets (e.g. Publix), grocery stores (e.g. Lexington Food Cooperative), convenience stores (e.g. 7-Eleven), "Other" (e.g. Whole Foods, Walgreens, independent grocers, etc.). Some gas stations (e.g. that are operated by Costco) are classified under "other" by USDA but are bigger than supermarkets (but the paper says that gas stations smaller than supermarkets were omitted). How are these classifications handled by authors? Other SNAP-eligible stores in the USDA dataset are small-scale (in footprint, number of employees, and sale) but do sell fresh fruits and vegetables - how would these be coded in the analysis? All this to say that the definition of "supermarket/grocery stores" is unclear in the paper, and, therefore, the analysis would be difficult to replicate with the information as provided.

3. Provide a link to the database in the paper (the link in Logan et al, 2022 is broken), and clarify year of data.

4. I would recommend that the authors briefly describe how the use of the Kolm-Pollak EDE index overcomes shortcomings of the GINI index that has been previously used to document inequitable food retailscapes (in Raja et al, 2008; Beyond Food Deserts, Journal of Planning Education and Research). One sentence would be plenty.

TAKEAWAYS

Notwithstanding the lack of clarity on the definition/use of supermarket data, I found the methodology used in the paper to be extremely useful for policy, if not in the way that the authors suggest. The authors suggest that the findings would be useful for urban planners. Unfortunately, urban planners in the United States have limited (if any) impact on the locational choices of the contemporary private sector food retail industry in the US (To understand the way in which urban planning and food systems interact read Clark et al 2021 Essential, Fragile, and Invisible Community Food Infrastructure; Raja et al 2018 Growing Food Connections through Urban Planning in Cabbanes, Yves (editor); Raja et al 2008 Beyond Food Deserts; and review/browse the growingfoodconnections.org local government food policy database). Briefly, some cities including Baltimore, NYC, Philadelphia, and Seattle have been proactive in influencing their food landscapes but even then neighborhood-scale food retail disparities persist, especially along racial lines. Even cities in the authors' data set which meet the 15-minute walking ideal have Black neighborhoods that are underserved by food retail (Chicago, for example). Urban planners can regulate what can happen on a piece of land (through zoning) but they have limited tools to attract new grocery, which tends to locate in inner-ring suburbs or exurbs. In Buffalo, NY, for example, the eastern third of the city, which is home to the majority of the city's Black population, has 1 major supermarket. Yet just outside the city borders - in the suburb of Amherst (about 5 km of the city) there are four+ supermarkets and supercenters (Whole Foods, Wegmans, Trader Joe's, Tops, Walmart, etc.), all clustered together. In other words, a real constraint for Buffalo's ability to become a 15-minute city is the fact that supermarkets choose to locate in the neighboring suburb located 20-minutes away. This is not a new problem - it has persisted for decades. Therefore, in my judgment, the authors findings would be extremely useful for state (and federal) governments to create/fund economic development incentives to level the playing field across metropolitan areas (city-suburb-exurb) rather than for urban planners to decide where food retail should be sited within cities.

Recommendations: Please speak to what the state/federal government can do with the findings (rather than only speaking to urban planners who have little/no control across a regional/metropolitan landscape). For example, the federal Healthy Food Financing Initiative (HFFI) might benefit hugely from the national-scale findings of this paper.

(Remarks on code availability)

No code or data file was available.

Reviewer #4

(Remarks to the Author)

This is an interesting manuscript that uses a newly developed approach to optimize the placement of supermarkets across the 500 largest American cities, in an effort to improve the walkability of cities. Their optimization model does not only place supermarkets based where there are no supermarkets available, rather their approach selects the optimal placement based on the premise of reducing inequalities within the local food environment. Overall, I really enjoyed reading this (this is very cool economic geography in practice), but I faced some challenges with the manuscript, primarily in terms of how inequality is really operationalized in the model and in terms of the food security theoretical underpinning of the paper. I'd like to see a more clear statement on what the inequality-minimizing approach really is, rather than asking the reader to wade through a separate (and unpublished) paper. Additionally, there needs to be a much stronger engagement with food access literature from the United States. Below, I have separated my comments into major and minor points. I look forward to seeing the next draft of this manuscript.

Major points:

1. At the end of the first sentence in the Abstract you state, "... but the practical implications of developing x-minute walkable neighborhoods have not been rigorously explored." I'm not sure this manuscript rigorously explores the practical implications, primarily due to the fact that you're stating more supermarkets are necessary to ensure walkable cities and equitable access to food. However, this assertion doesn't align with the very large pool of literature that has empirically shown food accessibility is not simply about availability (presence / absence) of supermarkets, nor about proximity (physical access) to supermarkets. There are a litany of reasons why people do (don't) shop at specific retailers. My main challenge is the assumption that adding supermarkets will alleviate food inaccessibility (and ultimately food security) and food-related equality issues. I understand that the manuscript is primarily about showing the power of this model, but because you've cited practicability as an important point, I'm hung up the notion that more supermarkets at such a scale are a practical solution - especially in terms of equity and justice challenges.

2. Below are some major comments related to other retailers:

- On L26, you state you evaluated access to grocery stores and supermarkets? There needs to be a much stronger justification for including only these two types of retailers. On Lines 47 to 49 you provide a brief statement, but this statement does not align well with the existing literature around food deserts (similar to the challenges I have with food accessibility lit). Essentially, the lit suggests that more supermarkets won't ameliorate food deserts, and also that food deserts is too binary of a term and obscures the on-the-ground reality of food environments, particularly in urban areas.

- On L30 you state, "we mean stores that sell food, including fresh produce, and are larger than convenience stores or gas stations." Why does the size matter? Is it because supermarkets are fewer in number; therefore, easier to map / include in the analysis? Or is it because past research finds these types of retailers typically offer less healthy options? If the latter, some citations are needed. If the former, that needs to be more clearly stated. Additionally, farmers markets and permanent brick and mortar farm stores are bigger than c-stores and gas stations. Why not include them?

- On L4, you mention that social cohesion stems from sustainable urban design. I'm surprised that public / farmers markets aren't included in this analysis. There's a massive amount of research on public markets contributing to intangible characteristics (including social cohesion) of communities.

- Research Question 1 (L36) states, "If a city can open..." In reality, cities and their governments can't open any supermarkets. They could possibly open farmers markets or incentivize supermarkets, but they can't open them.

3. The inequality measure / characteristics used to calculate the EDE feel like a black box. It's helpful to know there's a companion article, but at least a very brief summary would be useful here. Even simply stating what it is evaluating in terms of inequality in brief layman's terms would help the reader immensely. On top of that, I did look at your companion article and for someone not well-suited to highly computational texts, it was difficult to understand. I know that's an issue for me - less about your writing style - but I'm sure you want to ensure readability and clarity.

4. The manuscript, in general, doesn't substantively acknowledge on-going changes in agri-food value chains, where internet grocery shopping is common. Food access is no longer the result of in-person consumer-retailer interactions. Furthermore, Clapp's work around changing agri-food value chains also mentions that corporate consolidation of food systems is a major contributor to food-related inequalities. I see more and more supermarkets, like what you're suggesting, as a major contributor to consolidation issues. Can you speak to that?

5. It strikes me as odd that the paper is framed early on as being about equitability and sustainability in terms of transportation, but the empirical focus is on supermarket accessibility. For instance, Lines 170 to 172 are about transportation modes and choice. Why not shift the conceptualization to x-minute cities and their impact on food security / accessibility, rather than x-minute cities are about transportation and then seemingly stating that here's an example about supermarkets? I think the abstract is okay, but the first couple of paragraphs could (possibly) be more focused on the relationship between food access, inequalities, and proximity (x-min city).

6. I'm not sure the paragraph written from Lines 214 to 227 is necessary or helpful. The sentence describing how it could be applied to health and voting is fine, but the rest seems like it's repeated from elsewhere in the manuscript. Because so much of the manuscript is about food, this feels like an effort to tie it back in with the transportation framing and immediate proximity of urban amenities, but because so much of the paper is on food and food retailers, it feels awkward. I suggest you add that sentence and the one about the unique contribution to the paragraph below.

Minor points:

1. Line 2. What does the first sentence in the first paragraph actually mean - "The presence of amenities in urban areas..."? Do you mean, when there are amenities in an area, people are more likely to stay in / shop at those areas? As it is now, it isn't clear.

2. Line 8. You stated, "Cities worldwide are now articulating...". Can you provide specific examples of cities that are pursuing policies related to this? Please provide.

3. Table 1.

- The order of these cities isn't quite clear to me. I think you need to clearly state the cities are organized by the 20 lowest (best) EDE distances. That should be the second column, after the city column. As it is now, it takes too much mental dexterity to come to this relatively simple point.

- Is "metro population" the correct term to use here? I did a quick look and the metro area population for NYC is closer to 19 million (higher values for all of the other cities, as well).

- Please include a citation in the text stating that this data is from the 2020 US Census.

4. L49. This is just one of four ways that the USDA measures / evaluates food access. Also, importantly, food deserts aren't used as a term by USDA anymore. You need to more clearly justify why you've selected this definition.

5. Line 69 and 70. What does "Unfortunately, equity metrics tend to be algebraically complex, so optimization models that

contain them do not scale computationally to practical problem sizes” intuitively mean. Please restate this in a clearer manner.

6. L86, please be more specific on how your proposed placement leads to notable improvements for equality. In what way? Is it just that travel times are shorter? I assume it isn't that a larger population is served, because in Lines 59 to 72, you state that's a challenge with mean-minimizing models.

7. Lines 91 to 94. It appears that none of the Blocks have worse access (above the 1:1 line), is that correct? Am I interpreting this figure (Figure 1C) right? If I am incorrect, I'd amend the text to clarify how to interpret the graph.

8. You do a nice job later in the paper stating that you do this analysis at 5-, 10-, and 15-minute increments to satisfy variable desires by urban planners, but do you have any citations, data, quotes to show what cities actually want in terms of time ranges? It doesn't seem feasible to have supermarkets ~.5 minutes or even 10-minutes apart.

9. L116 and elsewhere in the manuscript. Because Miami features so heavily in the examples, why not include it in Table 1?

10. Lines 125 to 127. These numbers only add up to 161. What about the other 18 cities? $179 - 73 - 45 - 43 = 18$. Am I misunderstanding this sentence?

11. Can Figure 1 be moved up in the manuscript - closer to where you discuss the results in the text?

12. The units in the inset graph in Figure 1C are meters, right? Please indicate that.

13. I really like Figure 3!

14. Table 2. What does “average” indicate here? Average level of access? For instance, the average level of access for Seattle is 0km?

15. L170. Temporal variability of public transportation is also true of stores, no? Opening and closing times could also be incorporated? For more on this, look at

- Widener, M. J., & Shannon, J. (2014). When are food deserts? Integrating time into research on food accessibility. *Health & place*, 30, 1-3.

- Kaplan, K. H., Kirk, K. J., Lich, K. M., Palde, L. P. R., Van Allen, C., Nantz, E. L., ... & Knudsen, D. C. (2020). Accessibility to emergency food systems in south-central Indiana evaluated by spatiotemporal indices of pressure at county and pantry level. *Nature Food*, 1(5), 284-291.

16. L186. More is needed here. Cultural preferences, crime, etc. all shape who shapes where and when. Jerry Shannon's (Univ. of Georgia) work lays out a good deal of lit on why people shop where they do and when they do it.

- Shannon, J. (2016). Beyond the supermarket solution: Linking food deserts, neighborhood context, and everyday mobility. *Annals of the American Association of Geographers*, 106(1), 186-202.

17. On L244, please indicate that distance is Manhattan, not Euclidean.

18. Line 251. This makes me think you are using Manhattan distance, which is great (better than Euclidean, in this instance). What data is the Open Source Routing Machine using for the routes, and where does it come from?

(Remarks on code availability)

The authors indicate that the method (optimization model) that they use in the paper comes from work in another manuscript; however, it seems that other manuscript has not been published to this point. I don't feel comfortable with my skills / knowledge to comment on whether or not their code / quantitative methods are up-to-par or at the level required. However, I would feel more confident commenting on this paper, in general, if the paper they're pulling the method from was published at this point.

Version 1:

Reviewer comments:

Reviewer #1

(Remarks to the Author)

(Remarks on code availability)

After reading the revised version of the paper, we think the authors have done excellent work and have addressed most of our previous comments in a reasonable way. We still have a question: The authors added that their analysis focuses on Census Block Groups within the selected cities' administrative boundaries with densities greater than 200 housing units per km². However, this does not seem to influence the results presented in Figure 4, which are identical in the original and the

revised manuscript. Would it be possible to clarify why this is the case?

Reviewer #2

(Remarks to the Author)

(Remarks on code availability)

Reviewer #4

(Remarks to the Author)

Great job revising the text, both in terms of what the other reviewers and I mentioned, and in terms of general readability. Super work!

(Remarks on code availability)

Response to reviews - NCOMMS-24-27165-T

Dear reviewers,

Thank you for taking the time reviewing this work and your recommendations for improvement. We have responded to the specific comments and described our changes below. We have also revised the manuscript, in light of these changes, to maintain a clear flow.

Thank you again and happy new year.

Reviewer #1 (Remarks to the Author):

This work brings a very interesting approach to assessing the unequal distribution of opportunities within the urban space. The work builds on the Kolm-Pollak equally distributed equivalent (EDE) approach based on the Atkinson index. This is undoubtedly a novel approach to assessing spatial issues in urban systems. We believe this approach is promising for mapping spatial inequalities in cities (access to opportunities and exposure to hazards). However, it has conceptual weaknesses that can be overcome. Our purpose with this review is to help the authors find ways to overcome this challenge, or at least express the limitation of the analysis with more clarity and set the conceptual basis for advancing their ingenious method.

Many thanks for your time spent on this review and the thoughtful comments.

Conceptual challenges

The omission of urban density

The Atkinson index represents social welfare as an average utility. This is particularly relevant to the distribution of resources, such as income. The higher the mean income, the higher the social welfare; this is the idea behind it. Such an idea has been implemented through redistributive tax policies, and good evidence suggests it has been helpful in achieving more just and peaceful societies. However, the problem with cities and access to opportunities is different. It is fundamentally a spatial problem. The more centrally located one is in an urban system, the better the possibility of accessing more opportunities (see Space Syntax theory). The problem is that the lack of a central location cannot be easily compensated by subsidies, as is the case with income. Transport subsidies can alleviate the situation, but the disadvantage persists in the form of extra time (see Hägerstrand 1970 "What About People in Regional Science"). Indeed, in the last part of the abstract, the authors recognise that achieving the goal of a mean distribution of opportunities in contemporary cities is unfeasible. They claim it is because of a car-oriented urban design (which is part of the truth). However, premodern cities also had the same issue; people living in the periphery had a spatial disadvantage in accessing opportunities.

We have revised the manuscript to better reflect this fundamental spatial character of access and to clarify how our method addresses it:

- 1. We have updated our explanation of how the EDE specifically handles spatial distributions:** *"As illustrated in Figure 1, the EDE quantifies average access while penalizing greater distances, thereby prioritizing the needs of those with the poorest access. Unlike metrics that consider only distributional spread, such as the Gini index, which would prefer a solution where all residents are similarly far from an amenity over one where everyone is closer but with more variety in travel distances, the EDE incorporates both the average distance and its distribution."*
- 2. We acknowledge that spatial access is inherently uneven and that our goal is improvement rather than perfect equality:** *"While the x-minute city concept has gained popularity in urban planning, our method acknowledges that acceptable walking distances vary significantly across populations and contexts...By evaluating multiple distance thresholds, we demonstrate how the scale of intervention required varies with different accessibility targets. However, as we have argued previously {cite{Logan2022-mr}}, the ultimate goal should be improving accessibility rather than achieving arbitrary thresholds."*
- 3. We have reframed our discussion to focus on improving rather than equalizing access:** *"Unlike methods that would accept uniformly poor access to achieve equality, our approach identifies solutions that improve overall accessibility while reducing extreme inequalities. This finding suggests that cities can make their retrofit efforts more efficient by explicitly considering these distributional impacts when planning new amenity locations."*

In contemporary cities, the massification of cars has expanded accessibility to large peri-urban areas, decreasing urban density. The article acknowledges this issue (lines 162-3), but the analysis does not capture this crucial aspect. Many people choose to live in low-density areas because they believe access to more spacious housing, far from noisy, more compact urban cores, is worth paying with poorer access to opportunities (more distance, time, and transport costs). City planners know this is problematic, but it has been challenging to tackle in market-oriented societies where freedom of choice is highly regarded. However, making cities more compact and, therefore, more efficient remains a central endeavour for city planners. Taxpayers already subsidise this lifestyle in some way by maintaining roads and other elongated infrastructure at the state's expense (see, for example, the numerous studies addressing the subject of "the cost of sprawl"). Is this fair? In this light, what does "distributional justice accessibility" mean (line 215)? Can this, for example, be explained using Rawls' idea of justice as fairness (to mention one widely accepted definition of justice)? The authors claim in lines 178-80 that the model provides optimal equitable facility locations, enabling the transition to not only be efficient, but just. It would be interesting if the authors described their idea of justice and efficiency (just for whom and efficient for whom).

We have revised the manuscript to address these interconnected issues:

- 1. We now directly acknowledge the tension between density choices and infrastructure costs in the Discussion:** *"These challenges reflect a deep interdependence between urban design, sustainability, retail trends, and social equity. The consolidation of food retail into larger formats [Clapp2021-ny] both responds to and reinforces car-dependent urban form - as stores become larger and fewer, walking becomes less viable, which further increases car dependency... While some residents choose to live in lower-density areas, trading accessibility for other amenities, the infrastructure costs of serving dispersed development are often subsidized by the broader community."*
- 2. We have clarified our conceptualization of distributional justice by connecting it to established frameworks:** *"Distributional justice - how benefits and burdens should be distributed across society [Low2013-yx] - is central to this challenge. While geographic proximity represents just one dimension of food access, improving its distribution remains crucial for enabling transport choice and reducing disparities. Following Penchansky and Thomas' framework [Penchansky1981-qh], as extended by Saurman [Saurman2016-gj], comprehensive food access depends on multiple factors: availability, geographic accessibility, accommodation (e.g., store hours), affordability, acceptability (whether stores meet cultural preferences), and awareness."*
- 3. We make clear that our density threshold acknowledges economic realities while focusing improvements where they can be most effective:** *"Following the methodology of [Logan2022-mr], we focus our analysis on Census Block Groups within these cities' administrative boundaries with densities greater than 200 housing units per km². This density threshold is equivalent to moderate density in a suburban area and captures areas where walkable access is most feasible and infrastructure investments are most efficient. The analysis thus avoids recommending stores in locations where they would likely be economically unsustainable."*

Due to the low population density, many areas may not have grocery shops because the location is not economically feasible for the shop owner. Should the taxpayer then subsidise the location of non-economically feasible supermarkets in low-density areas because everybody should live within x minutes' distance from a grocery shop? It seems illogical to think this can be the author's idea, but if it is not the case, why was population density not part of the analysis?

As addressed in our previous responses, we explicitly incorporate density considerations through both our density threshold approach and population weighting in the EDE calculation. The analysis focuses on areas with sufficient density (>200 housing units per km²) to potentially support new stores, rather than suggesting taxpayer subsidies for stores in low-density areas.

The analysis has significant limitations without considering population density (or at least the number of served/non-served citizens). It presents a very coarse picture of underserved areas but does not provide an empirical base for the location of economically viable grocery stores. Such an analysis may require including different measures of urban density such as population, network density, floor space index, etc. Some areas with enough density to make grocery shops economically feasible may be, in fact, deprived of such service. If the authors can depict that, they are making a valuable contribution to planning authorities and helping grocery shop owners find economically feasible new locations. It can be challenging to include that for an analysis of 500 cities. Still, the limitation needs to be addressed more explicitly, explaining why density has not been possible to include in this current analysis. This inclusion can be an avenue to expand this research in the future. These limitations have been somehow addressed in section "Measuring Access" (lines 241-61), but they need to be more central in the argument. It is not a minor thing; density is fundamental for efficiency in urban areas.

We agree that density and economic viability are fundamental considerations. We have expanded our treatment of these issues in several ways:

- 1. As addressed in our previous responses, we incorporate density through both our threshold approach (>200 housing units per km²) and population weighting in the EDE calculation.**
- 2. We have expanded our Limitations section to explicitly address the need for additional density measures and economic considerations:** *"While we use a density threshold of 200 housing units per km² to focus on potentially viable locations, we do not consider economic factors like purchasing power, competition, or operating costs. Our method inherently weights access by population through the EDE calculation, favoring denser areas while ensuring equity through the inequality penalty. However, successful implementation would require considering market forces, community preferences, and political constraints beyond our analysis scope... Future work could also integrate other dimensions of access, analyze distributional impacts between socio-demographic groups [Logan2021-inequ], and incorporate additional measures of urban form and economic viability."*
- 3. We have clarified the practical application of our findings in the Discussion:** *"The practical application of these findings can inform multiple stakeholders working to improve food access across cities. Research underscores that successful initiatives require sustained, community-driven investment rather than relying on short-term interventions, which often fail to address structural inequities in food environments... Our optimization approach can contribute to these multi-faceted efforts by identifying priority areas where new food retail outlets would have the greatest impact on improving access. Our results should be interpreted as identifying areas of need and opportunity, rather than as direct recommendations for store placement."*

This revision makes clear that while our analysis provides a foundation for identifying promising locations, successful implementation requires considering additional measures of density and economic viability that were beyond the scope of this large-scale study.

There is a need to clarify the concept of city.

The sample selection seems to be based on the 500 most populous municipalities (cities, according to the paper). The challenge here is that cities are regional phenomena (see the concept of “functional urban area”), and exceptionally, access to opportunities is constrained by administrative boundaries. It seems the authors are aware of this phenomenon by including grocery store locations within a 5 km radius of the administrative boundaries of the cities in their analysis (lines 254-256). This is depicted in Figure 1, which illustrates that existing supermarkets outside the boundaries of Miami serve a part of Miami’s population. However, we recommend that authors clarify the definition of a city and its relationship with the phenomenon of functional urban area in the ‘Main’ section of the manuscript since the performed analysis is eventually constrained to the administrative jurisdiction of the examined cities. This will serve the narrative’s clarity and comprehension.

Thank you for this point of clarification. We have made several revisions to address this:

- 1. We have added a clear explanation of our city selection and boundaries in the Methods:** *"The cities analyzed in this study are those included in the Centers for Disease Control and Prevention (CDC) 500 Cities Project (superseded by their PLACES project [cdc_places]), which focused on the largest 500 U.S. cities. While the CDC project focused on chronic disease measures, our study repurposes this established set of cities to explore accessibility and equity in food retail distribution."*
- 2. We explicitly acknowledge the relationship between administrative and functional boundaries:** *"While our analysis focuses on cities defined by administrative boundaries, we recognize that urban areas often function as part of larger metropolitan systems. Our focus on administrative boundaries provides a standardized method for analyzing accessibility across multiple cities while acknowledging their connection to broader urban regions."*
- 3. We have clarified our treatment of cross-boundary interactions:** *"Although our analysis is constrained to the administrative boundaries of these cities, we account for aspects of the functional nature of urban systems by incorporating grocery store locations within a 5 km buffer zone beyond city limits. This approach acknowledges that urban areas extend beyond their formal boundaries, and amenities located just outside these limits may serve city residents."*

Minor comments

According to the proposed EDE optimisation model, the authors use Figure 3 and Table 2 to present the generated results regarding how many supermarkets are needed across the largest 500 US cities to achieve different access levels. While these figures provide a good overview for the reader, we recommend that alongside 'Average', it is also indicated that this category refers to the distance of '2290 m'. We believe that this will improve the comprehension of the figure and table.

We agree. Thank you for this suggestion.

Lines 165-172 elaborate on how public transportation usage can affect the generated results, but what about cycling? In specific cases, cycling can be an even more influential mobility choice with respect to accessibility to services than public transport. Therefore, we suggest articulating cycling as a potential mobility mode that can extend the studied amenities' catchment area.

We have noted the public transport and cycling in the Limitations section:

"Conversely, by focusing solely on walking access, we may overestimate required stores in cities where cycling infrastructure or public transport effectively extends catchment areas. While our optimization methodology could incorporate these alternative transport modes if travel time data were available, this study provides an initial estimate based on walking access."

Fabio Hernández Palacio and Todor Kesarovski

Thank you for your helpful feedback.

Reviewer #2 (Remarks to the Author):

Reviewer #2 (Remarks on code availability):

NCOMMS-24-27165-T

Thank you for taking the time to read and provide feedback.

Reviewer #3 (Remarks to the Author):

OVERVALL REFLECTION

The authors address a timely topic, use a deft methodological approach, and offer intriguing findings. That said, I do not entirely agree with the implications of the authors' findings, and I hope my reflections help the authors strengthen the paper (especially the discussion section). I recommend that the paper be published with minor revisions.

Many thanks to you for your constructive comments and encouragement to situate into the policy implications.

The authors investigate the degree to which cities can be retrofitted to achieve the so-called "15-minute city" ideal that has reappeared in popular discourse. Loosely speaking, a 15-minute city affords its residents the opportunity to reach life's amenities within a 15-minute walking/bicycling distance of their homes. The 15-minute city idea has been critiqued in the literature as lacking a theoretical framework and is extolled without sufficient empirical evidence for why and how such a vision might be implemented in cities. In this paper, the authors explore the feasibility of the 15-minute city idea by examining the residents' spatial access to a particular type of food retail in the 500 largest cities in the United States using the Kolm-Pollak equally-distributed equivalent (EDE) index. The use of the Kolm-Pollak equally-distributed equivalent (EDE) index is well executed, and illustrates a novel way of understanding inequitable access in a particular type of food retail across the US.

NOTEWORTHY RESULTS

The article, with the provocative title "Hundreds of supermarkets needed across the United States to achieve walkable cities," reports that 25% of the studied (500) cities in the US could, in theory, provide 15-minute access to supermarkets by adding fewer than 5 new supermarkets. The authors report that to reach the 15-minute ideal, cities in the study require on "average 18 additional supermarkets. [...] Some cities are not far from this target. 23 cities require only one supermarket and 62 require between two and five additional supermarkets. 106 cities require between six and ten additional supermarkets, while 132 require 11 to 20 more." Two look at two

examples from the author tables, NYC needs no supermarkets to become a 15-minute city while Jacksonville, Florida needs to add the most supermarkets (362) to become a 15-minute city.

Notwithstanding the provocative title of the article, the actual data from the article suggest that a relatively small number of additional supermarkets are necessary for many US cities to reach the 15-minute city ideal. Six of the 20 largest cities in the US - NYC, San Francisco, Philadelphia, Washington, Chicago, and Seattle - have already reached this quantitative/spatial ideal. As the authors note, 25% of cities in their dataset could reach this ideal by adding 5 or fewer supermarkets. Still, the findings are sobering - because it is extremely difficult to convince a supermarket proprietor to move (or, stay) in a neighborhood that they do not wish to be in for a variety of reasons.

We have updated the text in the Abstract and Discussion to state: “We found that 25% of cities could achieve 15-minute walking access by adding five or fewer stores in optimal locations, while more ambitious 5-minute targets would require over 100 additional stores in most cities.”

SIGNIFICANCE AND ORIGINALITY

This is a creative and original piece of work that adds empirical rigor to the idea of the 15-minute city. The methodological approach and findings are useful in illustrating that the 15-minute city ideal is already in place for some cities and out of reach for others (in so far as access to a particular type of food retail is concerned). The methodological approach can be adapted for each place gauge where food retail amenities (or, other amenities) might be added to create a more equitable food retailscape.

Recommendation: If you intend to speak to urban planners, consider reviewing/locating your work in urban planning literature that has documented the challenges in food retail (Raja et al, 2008), and how difficult it is to bring supermarkets into US cities (Kami Pothukuchi 2005). Consider the following journals: Economic Development Quarterly, Journal of the American Planning Association; Journal of Planning Education and Research; Journal of Agriculture, Food Systems, and Community Development; Environment and Planning.

Thank you for sharing these articles. We've updated the Introduction to better reflect the stakeholders that this work could support:

“The answers to these questions provide crucial insights about the scale of change required to retrofit car-oriented cities for both improved equity and sustainability. While cities cannot directly open supermarkets, research has shown that cities with successful grocery initiatives are those that have shown political leadership, engaged with public agencies, collaborated across jurisdictions, and partnered with community-based nonprofits and existing small-scale food retailers \cite{Pothukuchi2005-no, Raja2008-wx, Raja2018-xo}.

This analysis therefore can inform multiple stakeholders: local governments using zoning and incentives, state and federal agencies developing funding programs like the Healthy Food Financing Initiative, retailers identifying promising locations, and community

organizations advocating for improved food access. Our optimization approach provides a rigorous, equity-focused method to guide these decisions across different urban contexts."

We have also significantly updated the Discussion to reflect on these challenges and the implications of our work. We provide more detail in response to your other comments.

THEORETICAL FRAMING

The authors reference the presence of "food deserts" in the United States to justify the significance of their work. This, in my judgment, undermines their own excellent focus (and analysis) on equitable access. The food deserts literature has been heavily criticized for its lack of theoretical framing, weak empirical analyses, and deficit-based view of Black and brown communities in the United States. In fact, the authors' use of the Kolm-Pollak EDE, which focuses on equity (not equality), is a welcome departure from the food deserts literature.

Recommendation: For stronger theoretical framing, I recommend reviewing the extensive food apartheid literature and the literature on white accumulation which explains why amenities are located just outside of city boundaries (in inner-ring suburbs). Edward Goetz writes about this in the context of housing, for example. See this short piece by Food Equity scholars which highlights how food apartheid and accumulation work hand-in-glove in food retail disparities (<https://civileats.com/2022/05/24/op-ed-east-buffalo-needs-community-driven-structural-investments-not-fly-in-fly-out-charity/>). Frankly, I would be delighted to see the analysis of this paper executed for metropolitan areas (rather than cities) because most food retail inequity is more pronounced across the city-suburb scale than within cities [The authors are already getting at this a bit because they have included supermarkets within 5 km boundary of cities in their analysis, but the theoretical reasoning for doing so is absent/weak.]

Thank you. We understand that the "food desert" framing was problematic and have revised our manuscript to better reflect the structural causes of food inequities. We have:

- 1. Removed references to "food deserts" and instead framed our work within broader patterns of environmental injustice and structural inequity in urban environments.**
- 2. Added discussion of how spatial patterns of supermarket access reflect historical disinvestment and discriminatory policies, citing Food Equity Scholars (2022):** *" These disparities reflect historical patterns of disinvestment and discriminatory policies, with supermarkets often concentrated in wealthier, predominantly white suburbs; Rather than being naturally occurring phenomena, these spatial patterns result from deliberate policy choices that have created and reinforced inequitable food environments."*

3. **Better explained our theoretical approach by incorporating Penchansky and Thomas' framework of access dimensions, acknowledging that geographic proximity is just one aspect of a complex food environment that includes availability, accommodation, affordability, acceptability, and awareness:** *"Following Penchansky and Thomas' framework \cite{Penchansky1981-qh}, as extended by Saurman \cite{Saurman2016-gj}, comprehensive food access depends on multiple factors: availability, geographic accessibility, accommodation (e.g., store hours), affordability, acceptability (whether stores meet cultural preferences), and awareness. Our analysis focuses on geographic accessibility while acknowledging these broader dimensions."*
This theme of dimensions of access also is included in the limitations and the discussion.
4. **Clarified our methodological choice to include stores within a 5km buffer of city boundaries:** *"Although our analysis is constrained to the administrative boundaries of these cities, we account for aspects of the functional nature of urban systems by incorporating grocery store locations within a 5 km buffer zone beyond city limits. This approach acknowledges that urban areas extend beyond their formal boundaries, and amenities located just outside these limits may serve city residents."*

While expanding the analysis to full metropolitan areas would be valuable future work, our current focus on administrative cities with their 5km buffers provides a standardized method for analyzing accessibility across multiple cities while acknowledging their connection to broader urban regions. The choice of administrative boundaries also aligns with the jurisdiction of local planning authorities, though we emphasize that addressing food access inequities requires coordination across multiple scales of government.

METHODS AND ANALYSIS

As noted earlier in this review, the use of the Kolm-Pollak equally-distributed equivalent (EDE) index is clever. Empirical work of the kind offered in this paper is much needed especially because (some) government officials fail to interrogate what reaching a 15-minute ideal might mean in a particular urban-regional context.

Recommendations for improvement

On page 5, the authors write that "we use the words grocery stores and supermarkets interchangeably; by both we mean stores that sell food, including fresh produce, and are larger than convenience stores or gas stations". The authors also point to an earlier article (Logan et al, 2022) for their definitions/use of supermarket data. Additionally, the authors note (on page. 17), "we use existing supermarket locations within a 5km radius of the city from the USDA's Food and Nutrient Service SNAP database available on ArcGIS Hub." The available information does not fully explain the type of store that the authors have included in the analysis - and this matters a great deal for the interpretation/discussion of their paper. Presuming that the analysis

is based on the USDA database that provides locations of all stores that are authorized to accept Supplemental Nutrition Assistance Program (SNAP) benefits, I would recommend the following clarifications:

1. Clarify that the paper is based only on retail stores that accept SNAP benefits (not all supermarkets/grocery stores; there are some supermarkets/grocery stores that may not accept SNAP benefits). In my judgment this is not a problem - in fact, if there are stores that do not accept SNAP they are by definition inaccessible to people who depend on SNAP benefits. The authors simply need to say this.

2. Additionally, I would recommend that the authors report precisely the type of SNAP food retail that they have included in their analysis. As I understand it, the USDA SNAP retail location database includes multiple categories: superstores (e.g. Walmart, an increasingly common source of fresh produce for low-income people), supermarkets (e.g. Publix), grocery stores (e.g. Lexington Food Cooperative), convenience stores (e.g. 7-Eleven), "Other" (e.g. Whole Foods, Walgreens, independent grocers, etc.). Some gas stations (e.g. that are operated by Costco) are classified under "other" by USDA but are bigger than supermarkets (but the paper says that gas stations smaller than supermarkets were omitted). How are these classifications handled by authors? Other SNAP-eligible stores in the USDA dataset are small-scale (in footprint, number of employees, and sale) but do sell fresh fruits and vegetables - how would these be coded in the analysis? All this to say that the definition of "supermarket/grocery stores" is unclear in the paper, and, therefore, the analysis would be difficult to replicate with the information as provided.

3. Provide a link to the database in the paper (the link in Logan et al, 2022 is broken), and clarify year of data.

Thank you for raising these three points. We have:

- 1. Added explicit acknowledgment that our analysis uses SNAP-authorized retailers from the USDA database. As you note, while this might exclude some stores that don't accept SNAP benefits, focusing on SNAP-authorized retailers ensures we capture stores that are accessible to residents who depend on these benefits.**
- 2. Updated the Introduction to state this, clearly define supermarket, and acknowledge the role of smaller food outlets:**

"We use supermarket locations from the USDA's Supplemental Nutrition Assistance Program (SNAP) retailer database, defining supermarkets as full-service stores that typically offer a broad range of groceries including fresh produce, meats, and deli items \cite{Kolak2018-az}. \autoref{tab:cities_assess} shows the largest 20 cities in the study, along with their residents' inequality-penalized average access and their ranking among all 500 cities. While food environments are complex and include many other important sources such as farmers markets, small grocers, and specialty stores, we focus on supermarkets due to data availability and consistency across all 500 cities in our study. When applied at a local scale, our method can optimize locations for other facility types,

or multiple facility types simultaneously, given the critical role of smaller food outlets for improving food access in minority neighborhoods \cite{Raja2008}.”

3. Expanded our explanation of how we classified supermarkets in the Methods and added the date and URL:

“We use existing supermarket locations from the USDA's Food and Nutrient Service 2021 Supplemental Nutrition Assistance Program (SNAP) database \cite{usda_snap_retailer_2021}, which catalogs retailers authorized to accept SNAP benefits. Although the 2021 data is no longer available, more recent data is available at <https://usda-fns.hub.arcgis.com>. Following Kolak et al. \cite{Kolak2018-az}, we define supermarkets as full-service stores that typically offer a broad range of groceries, including fresh produce, meats, and deli items. The store should have five or more checkout lanes \cite{Kolak2018-az}. The criteria were chosen to include stores offering a variety of healthy food options \cite{Block2006-dd}. We identified supermarkets using a list of national and regional chains that we compiled, explicitly excluding convenience stores, gas stations, and smaller retail formats. While this approach may exclude some smaller stores that sell fresh produce, it provides a classification method that can be applied at scale across the 500 cities. Our focus on SNAP-authorized retailers ensures we capture stores that are accessible to residents who depend on these benefits.”

4. I would recommend that the authors briefly describe how the use of the Kolm-Pollak EDE index overcomes shortcomings of the GINI index that has been previously used to document inequitable food retailscapes (in Raja et al, 2008; Beyond Food Deserts, Journal of Planning Education and Research). One sentence would be plenty.

We have updated our explanation of why the EDE is particularly suited for evaluating spatial access:

“As illustrated in \autoref{fig:what_ede}, the EDE quantifies average access while penalizing greater distances, thereby prioritizing the needs of those with the poorest access. Unlike metrics that consider only distributional spread, such as the Gini index, which would prefer a solution where all residents are similarly far from an amenity over one where everyone is closer but with more variety in travel distances, the EDE incorporates both the average distance and its distribution. For example, if most residents live within a 10-minute walk of a store but some must walk 30 minutes, the EDE would exceed the simple average to reflect this inequality. This makes it particularly suited for evaluating urban access, as it helps planners identify solutions that improve overall accessibility while reducing inequalities.”

Also, in the Discussion: *“While previous approaches using metrics like the Gini index focus only on reducing inequality, potentially at the cost of worse average access, our EDE-based optimization improves both the average distance and its distribution.”*

TAKEAWAYS

Notwithstanding the lack of clarity on the definition/use of supermarket data, I found the methodology used in the paper to be extremely useful for policy, if not in the way that the authors suggest. The authors suggest that the findings would be useful for urban planners. Unfortunately, urban planners in the United States have limited (if any) impact on the locational choices of the contemporary private sector food retail industry in the US (To understand the way in which urban planning and food systems interact read Clark et al 2021 Essential, Fragile, and Invisible Community Food Infrastructure; Raja et al 2018 Growing Food Connections through Urban Planning in Cabbanes, Yves (editor); Raja et al 2008 Beyond Food Deserts; and review/browse the growingfoodconnections.org local government food policy database). Briefly, some cities including Baltimore, NYC, Philadelphia, and Seattle have been proactive in influencing their food landscapes but even then neighborhood-scale food retail disparities persist, especially along racial lines. Even cities in the authors' data set which meet the 15-minute walking ideal have Black neighborhoods that are underserved by food retail (Chicago, for example). Urban planners can regulate what can happen on a piece of land (through zoning) but they have limited tools to attract new grocery, which tends to locate in inner-ring suburbs or exurbs. In Buffalo, NY, for example, the eastern third of the city, which is home to the majority of the city's Black population, has 1 major supermarket. Yet just outside the city borders - in the suburb of Amherst (about 5 km of the city) there are four+ supermarkets and supercenters (Whole Foods, Wegmans, Trader Joe's, Tops, Walmart, etc.), all clustered together. In other words, a real constraint for Buffalo's ability to become a 15-minute city is the fact that supermarkets choose to locate in the neighboring suburb located 20-minutes away. This is not a new problem - it has persisted for decades. Therefore, in my judgment, the authors findings would be extremely useful for state (and federal) governments to create/fund economic development incentives to level the playing field across metropolitan areas (city-suburb-exurb) rather than for urban planners to decide where food retail should be sited within cities.

Recommendations: Please speak to what the state/federal government can do with the findings (rather than only speaking to urban planners who have little/no control across a regional/metropolitan landscape). For example, the federal Healthy Food Financing Initiative (HFFI) might benefit hugely from the national-scale findings of this paper.

Thank you for this wider context. We have updated the Discussion to enable us to better incorporate the implications of the work: *"The practical application of these findings can inform multiple stakeholders working to improve food access across cities. Research underscores that successful initiatives require sustained, community-driven investment rather than relying on short-term interventions, which often fail to address structural inequities in food environments \cite{Food-Equity-Scholars2022-yo, Pothukuchi2016-tk}. While cities can foster supportive environments through comprehensive planning frameworks and zoning reforms, achieving long-term success requires addressing persistent challenges such as the perceived risks and complexities of urban development \cite{Pothukuchi2005-no}, the need for regional and cross-jurisdictional collaboration to bridge uneven capacities across municipalities \cite{Raja2018-xo, Clark2021-jn}, and the importance of strengthening local food infrastructure through targeted policies and investment*

\cite{Dillemath2016-kz}. Focusing exclusively on attracting large supermarkets risks overlooking opportunities to enhance food access through existing small-scale retailers, which often serve minority and low-income communities \cite{Raja2008-wx, Pothukuchi2016-tk}. Evidence from Detroit FRESH and other initiatives highlights that meaningful community engagement, combined with long-term financial and policy support, is critical for the success of these efforts \cite{Pothukuchi2016-tk, Food-Equity-Scholars2022-yo}.

Our optimization approach can contribute to these multi-faceted efforts by identifying priority areas where new food retail outlets would have the greatest impact on improving access. Our results should be interpreted as identifying areas of need and opportunity, rather than as direct recommendations for store placement. Local governments could use this model to guide targeted incentive programs, while state and federal agencies might allocate funding—such as through the Healthy Food Financing Initiative (HFFI)—to underserved areas \cite{Raja2018-xo, Dillemath2016-kz}. Community organizations could advocate for supermarket placements that align with neighborhood needs, fostering collaboration between public and private sectors \cite{Clark2021-jn, Pothukuchi2005-no}. Importantly, achieving lasting improvements requires moving beyond disconnected solutions, such as the addition of isolated stores, to comprehensive strategies that combine optimizing new supermarket locations with investments in existing local food retailers, developing robust regional food infrastructure, and ensuring community leadership in decision-making processes \cite{Raja2018-xo, Food-Equity-Scholars2022-yo, Clark2021-jn}.”

Additionally, we briefly introduce this in the introduction:

“The answers to these questions provide crucial insights about the scale of change required to retrofit car-oriented cities for both improved equity and sustainability. While cities cannot directly open supermarkets, research has shown that cities with successful grocery initiatives are those that have shown political leadership, engaged with public agencies, and partnered with community-based nonprofits and existing small-scale food retailers \cite{Pothukuchi2005-no, Raja2008}. This analysis therefore can inform multiple stakeholders: local governments using zoning and incentives, state and federal agencies developing funding programs like the Healthy Food Financing Initiative, retailers identifying promising locations, and community organizations advocating for improved food access. Our optimization approach provides a rigorous, equity-focused method to guide these decisions across different urban contexts.”

Reviewer #3 (Remarks on code availability):

No code or data file was available.

**We have updated the paper with a public GitHub link:
https://github.com/drewhort/equitable_facility_location**

Reviewer #4 (Remarks to the Author):

This is an interesting manuscript that uses a newly developed approach to optimize the placement of supermarkets across the 500 largest American cities, in an effort to improve the walkability of cities. Their optimization model does not only place supermarkets based where there are no supermarkets available, rather their approach selects the optimal placement based on the premise of reducing inequalities within the local food environment. Overall, I really enjoyed reading this (this is very cool economic geography in practice), but I faced some challenges with the manuscript, primarily in terms of how inequality is really operationalized in the model and in terms of the food security theoretical underpinning of the paper. I'd like to see a more clear statement on what the inequality-minimizing approach really is, rather than asking the reader to wade through a separate (and unpublished) paper. Additionally, there needs to be a much stronger engagement with food access literature from the United States. Below, I have separated my comments into major and minor points. I look forward to seeing the next draft of this manuscript.

Thank you for your detailed and constructive comments.

Major points:

1. At the end of the first sentence in the Abstract you state, "... but the practical implications of developing x-minute walkable neighborhoods have not been rigorously explored." I'm not sure this manuscript rigorously explores the practical implications, primarily due to the fact that you're stating more supermarkets are necessary to ensure walkable cities and equitable access to food. However, this assertion doesn't align with the very large pool of literature that has empirically shown food accessibility is not simply about availability (presence / absence) of supermarkets, nor about proximity (physical access) to supermarkets. There are a litany of reasons why people do (don't) shop at specific retailers. My main challenge is the assumption that adding supermarkets will alleviate food inaccessibility (and ultimately food security) and food-related equality issues. I understand that the manuscript is primarily about showing the power of this model, but because you've cited practicability as an important point, I'm hung up on the notion that more supermarkets at such a scale are a practical solution - especially in terms of equitability and justice challenges.

Thank you for this point. We too understand that food access is multifaceted and cannot be reduced to just geographic proximity and have made revisions throughout the paper to avoid giving this impression. We will detail these further in response to your other comments, but they include:

- 1. Better contextualize our focus on geographic accessibility within Penchansky and Thomas' comprehensive framework of access dimensions (availability, geographic accessibility, accommodation, affordability, acceptability, and awareness).**

2. Clarify that while improving geographic access alone cannot solve food inequities, it remains an important component for enabling transport choice and reducing spatial disparities.
3. Update our framing to emphasize that our method provides a tool for optimizing facility locations while considering equity, rather than suggesting that adding stores alone will solve food access challenges.
4. Acknowledge in our Discussion that successful initiatives require sustained, community-driven investment rather than relying on short-term interventions. We now explain how our optimization approach can contribute to these broader efforts by helping identify priority areas for intervention, while recognizing that implementation requires considering multiple stakeholders and factors beyond just proximity.

We have revised the Abstract's opening to better reflect this nuanced understanding of the challenge and our specific contribution to addressing one component of it: *“The location of amenities in urban areas fundamentally shapes both sustainability and equity outcomes. While cities worldwide are pursuing walkable neighborhood initiatives, the practical implications of retrofitting existing urban areas remain unclear. How many new facilities are needed, and where should they be located to ensure equitable access? We...These findings demonstrate both the potential for strategic interventions to efficiently improve access and the significant challenge posed by car-oriented urban development. By identifying priority areas for new facilities while considering distributional impacts, our method can inform multiple stakeholders working to create more sustainable and equitable cities - from local governments using zoning and incentives to state agencies developing funding programs and community organizations advocating for improved food access.”*

2. Below are some major comments related to other retailers:

- On L26, you state you evaluated access to grocery stores and supermarkets? There needs to be a much stronger justification for including only these two types of retailers. On Lines 47 to 49 you provide a brief statement, but this statement does not align well with the existing literature around food deserts (similar to the challenges I have with food accessibility lit). Essentially, the lit suggests that more supermarkets won't ameliorate food deserts, and also that food deserts is too binary of a term and obscures the on-the-ground reality of food environments, particularly in urban areas.

- On L30 you state, “we mean stores that sell food, including fresh produce, and are larger than convenience stores or gas stations.” Why does the size matter? Is it because supermarkets are fewer in number; therefore, easier to map / include in the analysis? Or is it because past research finds these types of retailers typically offer less healthy options? If the latter, some citations are needed. If the former, that needs to be more clearly stated. Additionally, farmers markets and permanent brick and mortar farm stores are bigger than c-stores and gas stations. Why not include them?

- On L4, you mention that social cohesion stems from sustainable urban design. I'm surprised that public / farmers markets aren't included in this analysis. There's a massive amount of research on public markets contributing to intangible characteristics (including social cohesion) of communities.

Thank you. We have substantially revised the manuscript to address these concerns. Our revisions acknowledge both the practical reasons for our focus on supermarkets (data consistency across 500 cities) while recognizing the important role of other food retailers in creating comprehensive food environments. We also emphasize that our method can be adapted to include multiple facility types when applied at local scales where more comprehensive data is available.

- 1. Regarding justification for focusing on supermarkets: We have clarified our approach and its limitations in the Methods section:** *"We use existing supermarket locations from the USDA's Food and Nutrient Service 2021 Supplemental Nutrition Assistance Program (SNAP) database [usda_snap_retailer_2021], which catalogs retailers authorized to accept SNAP benefits. Following Kolak et al. [Kolak2018-az], we define supermarkets as full-service stores that typically offer a broad range of groceries, including fresh produce, meats, and deli items. The store should have five or more checkout lanes [Kolak2018-az]. The criteria were chosen to include stores offering a variety of healthy food options [Block2006-dd]. We identified supermarkets using a comprehensive list of national and regional chains, explicitly excluding convenience stores, gas stations, and smaller retail formats."*
- 2. Regarding other food retailers: We acknowledge their importance throughout the manuscript:** *"While food environments are complex and include many other important sources such as farmers markets, small grocers, and specialty stores, we focus on supermarkets due to data availability and consistency across all 500 cities in our study. When applied at a local scale, our method can optimize locations for other facility types, or multiple facility types simultaneously, given the critical role of smaller food outlets for improving food access in minority neighborhoods [Raja2008-wx]."*
- 3. Regarding implementation: We have expanded our discussion of how this work can support comprehensive food access strategies:** *"Our optimization approach can contribute to these multi-faceted efforts by identifying priority areas where new food retail outlets would have the greatest impact on improving access. Our results should be interpreted as identifying areas of need and opportunity, rather than as direct recommendations for store placement... Focusing exclusively on attracting large supermarkets risks overlooking opportunities to enhance food access through existing small-scale retailers, which often serve minority and low-income communities [Raja2008-wx, Pothukuchi2016-tk]."*

- 4. Regarding methodology limitations: We have added to the Limitations section:** *"Our focus on SNAP-authorized supermarkets provides only a partial view of food environments. While these retailers typically offer broad food options [Kolak2018-az], many communities are also served by farmers markets, specialty stores, and smaller grocers that our analysis excludes... Our method could incorporate these additional retailers and temporal patterns if consistent data were available across all cities."*

- Research Question 1 (L36) states, "If a city can open..." In reality, cities and their governments can't open any supermarkets. They could possibly open farmers markets or incentivize supermarkets, but they can't open them.

Thank you. In our wider revisions we have clearly identified the relevant stakeholders who could be informed by this information. In response to this comment, we've updated the question to: *"If a city can encourage the opening of additional supermarkets, where should they be located to best improve equitable access?"*

3. The inequality measure / characteristics used to calculate the EDE feel like a black box. It's helpful to know there's a companion article, but at least a very brief summary would be useful here. Even simply stating what it is evaluating in terms of inequality in brief layman's terms would help the reader immensely. On top of that, I did look at your companion article and for someone not well-suited to highly computational texts, it was difficult to understand. I know that's an issue for me - less about your writing style - but I'm sure you want to ensure readability and clarity.

We agree that making this concept more accessible to readers is important. To help readers understand how the EDE works we have added Figure 1, which visually illustrates how the EDE relates to a distribution of access distances and the mean, making the concept more intuitive.

Also, we have expanded our explanation in the main text to include both conceptual and practical descriptions of how the EDE works: *"As illustrated in Figure 1, the EDE quantifies average access while penalizing greater distances, thereby prioritizing the needs of those with the poorest access. Unlike metrics that consider only distributional spread, such as the Gini index, which would prefer a solution where all residents are similarly far from an amenity over one where everyone is closer but with more variety in travel distances, the EDE incorporates both the average distance and its distribution. For example, if most residents live within a 10-minute walk of a store but some must walk 30 minutes, the EDE would exceed the simple average to reflect this inequality. This makes it particularly suited for evaluating urban access, as it helps planners identify solutions that improve overall accessibility while reducing inequalities."*

4. The manuscript, in general, doesn't substantively acknowledge on-going changes in agri-food value chains, where internet grocery shopping is common. Food access is no longer the result of in-person consumer-retailer interactions. Furthermore, Clapp's work around changing agri-food value chains also mentions that corporate consolidation of food systems is a major contributor to food-related inequalities. I see more and more supermarkets, like what you're suggesting, as a major contributor to consolidation issues. Can you speak to that?

This speaks to the interdependent challenges between urban design, sustainability, and equity. We have included the following point in the Discussion:

"These complexities reflect a deep interdependence between urban design, sustainability, retail trends, and social equity. The consolidation of food retail into larger formats \cite{Clapp2021-ny} both responds to and reinforces car-dependent urban form - as stores become larger and fewer, walking becomes less viable, which further increases car dependency. This pattern particularly affects communities that already face multiple barriers to food access. Our findings about the number of stores needed to achieve walkable access underscore the scale of this challenge - the substantial investment required in many cities reflects decades of car-oriented development and retail consolidation. Breaking this cycle requires coordinated action across scales, from neighborhood-level support for smaller food retailers to regional policies that address broader patterns of retail distribution. While improving walkable access alone cannot solve food inequities, it represents an important step toward both more sustainable and more equitable urban environments."

5. It strikes me as odd that the paper is framed early on as being about equitability and sustainability in terms of transportation, but the empirical focus is on supermarket accessibility. For instance, Lines 170 to 172 are about transportation modes and choice. Why not shift the conceptualization to x-minute cities and their impact on food security / accessibility, rather than x-minute cities are about transportation and then seemingly stating that here's an example about supermarkets? I think the abstract is okay, but the first couple of paragraphs could (possibly) be more focused on the relationship between food access, inequalities, and proximity (x-min city).

Thank you. We have substantially revised the opening paragraphs to center food access and equity rather than transportation. The revised introduction now begins:

"The location of amenities in urban areas fundamentally shapes both sustainability and equity outcomes. While walkable access to daily needs offers multiple benefits, many cities' car-oriented urban design limits these opportunities [Logan2022-mr, Wu2021-nx]. Between 1990 and 2014, urban sprawl increased globally by 95% [Behnisch2022-xa], creating environments where essential services like grocery stores remain inaccessible without a car. These design choices disproportionately affect minority and low-income communities [Kolak2018-az, Raja2008-wx], compounding existing disparities in health

outcomes and quality of life [Leyden2003-dg, Lopez2011-cc]. Food access is critical to this challenge - despite being essential for wellbeing, grocery stores in many U.S. cities remain car-dependent and inequitably distributed, with minority and low-income communities facing systematically greater barriers to access..."

6. I'm not sure the paragraph written from Lines 214 to 227 is necessary or helpful. The sentence describing how it could be applied to health and voting is fine, but the rest seems like its repeated from elsewhere in the manuscript. Because so much of the manuscript is about food, this feels like an effort to tie it back in with the transportation framing and immediate proximity of urban amenities, but because so much of the paper is on food and food retailers, it feels awkward. I suggest you add that sentence and the one about the unique contribution to the paragraph below.

This paragraph has been removed as part of the wider revision.

Minor points:

1. Line 2. What does the first sentence in the first paragraph actually mean - "The presence of amenities in urban areas..."? Do you mean, when there are amenities in an area, people are more likely to stay in / shop at those areas? As it is now, it isn't clear.

As part of the reframing you recommended, this line has been removed.

2. Line 8. You stated, "Cities worldwide are now articulating...". Can you provide specific examples of cities that are pursuing policies related to this? Please provide.

We have revised this paragraph in the introduction: *"Cities worldwide are now articulating visions of improving their residents' proximity to amenities and thereby capitalizing on the benefits of active transport \cite{C40_Cities2020-bm}. These range from ambitious 5-minute targets in Copenhagen to 20-minute goals in cities like Portland and Glasgow, with many cities including Paris, Ottawa, and Shanghai adopting 15-minute targets. While car access has expanded mobility for many, true transport freedom requires that walking remains viable for daily needs. This is particularly important for grocery shopping - research in the Netherlands found that 50\% of respondents consider 500 meters (approximately a 5-6 minute walk) too far to carry groceries \cite{Schaap2016-ov}. This sustainable transition raises significant questions for our existing cities about how they can direct this retrofit efficiently and effectively."*

3. Table 1.

- The order of these cities isn't quite clear to me. I think you need to clearly state the cities are organized by the 20 lowest (best) EDE distances. That should be the second column, after the city column. As it is now, it takes too much mental dexterity to come to this relatively simple point.

We changed the order of the 2nd and 3rd columns and clarified the text of Table 1.

- Is "metro population" the correct term to use here? I did a quick look and the metro area population for NYC is closer to 19 million (higher values for all of the other cities, as well).

This is urban population based on the administrative boundary and the urban areas. "Metro Population" has been renamed "Urban Population".

- Please include a citation in the text stating that this data is from the 2020 US Census.

This is now cited in the table.

4. L49. This is just one of four ways that the USDA measures / evaluates food access. Also, importantly, food deserts aren't used as a term by USDA anymore. You need to more clearly justify why you've selected this definition.

As part of the reframing, we have removed references to "food deserts" and instead frame our work within broader patterns of environmental injustice and structural inequity in urban environments. E.g.,:

"The inequitable distribution of food retailers exemplifies broader patterns of environmental injustice, where disadvantaged populations systematically face greater barriers to accessing essential services \cite{White2021-jd, Fussel2010-te, Bulkeley2014-so}. These disparities reflect historical patterns of disinvestment and discriminatory policies, with supermarkets often concentrated in wealthier, predominantly white suburbs; Rather than being naturally occurring phenomena, these spatial patterns result from deliberate policy choices that have created and reinforced inequitable food environments \cite{Food-Equity-Scholars2022-yo}."

5. Line 69 and 70. What does "Unfortunately, equity metrics tend to be algebraically complex, so optimization models that contain them do not scale computationally to practical problem sizes" intuitively mean. Please restate this in a clearer manner.

Thank you for raising this. The explanation has been shifted to the Methods and revised to:

"Many models aimed at incorporating equity in facility location optimization have been proposed \cite{MARSH19941, Smith2013-md, Batta2014-uj, MARSH19941, Xu2023-iy, Karsu2015-cb}. However, the computational expense required to solve these models has limited their practical value. Modern optimization solvers can handle either nonlinear functions or integer-valued variables in large problems, but not both simultaneously. Since facility location models require integer variables (0/1-valued variables that indicate whether to place a facility at a potential location), including a nonlinear equity metric severely restricts the size of problems that can be solved \cite{Barbati2018-nd}."

6. L86, please be more specific on how your proposed placement leads to notable improvements for equality. In what way? Is it just that travel times are shorter? I assume it isn't that a larger population is served, because in Lines 59 to 72, you state that's a challenge with mean-minimizing models.

We added the following to clarify the difference between equality-optimizing and mean-optimizing (traditional) approaches. This includes elaborating what was L86: *"... our proposed placement leads to notable improvements for equality vs the traditional approach. In particular, there are often competing decisions of whether to choose a location that improves access by a few meters for many residents who already have good access, or to choose a location that dramatically improves access for fewer residents who currently have very poor access. When both of these solutions result in the same improvement in mean distance, our model chooses the latter, selecting the solution that improves the situation of those who are currently more disadvantaged."*

We also clarified how to interpret Figure 1C: *"The further below the 1:1 line a point is, the more the corresponding Block's access has improved. No Block's access has gotten worse because we are only adding supermarket locations, not taking them away. This figure shows the distribution effect because it shows which Blocks experience the greatest improvement in access under each intervention. By comparing the traditional vs inequality-minimizing approaches, we see that the approach that seeks to optimize inequality tends to improve the access of currently access-poor areas in comparison to the traditional approach."*

7. Lines 91 to 94. It appears that none of the Blocks have worse access (above the 1:1 line), is that correct? Am I interpreting this figure (Figure 1C) right? If I am incorrect, I'd amend the text to clarify how to interpret the graph.

You are correct, our response to your point above includes how we clarified this wording.

8. You do a nice job later in the paper stating that you do this analysis at 5-, 10-, and 15-minute increments to satisfy variable desires by urban planners, but do you have any do you have any citations, data, quotes to show what cities actually want in terms of time ranges? It doesn't seem feasible to have supermarkets ~.5 minutes or even 10-minutes apart.

Yes. Firstly, we have listed some of the cities that have set targets for accessibility: *"These range from ambitious 5-minute targets in Copenhagen to 20-minute goals in cities like Portland and Glasgow, with many cities including Paris, Ottawa, and Shanghai adopting 15-minute targets."*

Additionally, we refer to two surveys of preference for accessibility: *"While the x\$-minute city concept has gained popularity in urban planning, our method*

acknowledges that acceptable walking distances vary significantly across populations and contexts. Research shows this variability clearly - in the Netherlands, only 50% of respondents considered 400 meters an acceptable distance to walk with groceries \cite{Schaap2016-ov}, while a New Zealand study found that willingness to travel to supermarkets varies by time of day, gender, age, and ability \cite{White2024-hg}.

9. L116 and elsewhere in the manuscript. Because Miami features so heavily in the examples, why not include it in Table 1?

Miami has been added to the table and the caption updated to say “Miami (population: 441,228) is included due to its use as an example case throughout the paper, though it is not among the 20 largest cities.”

10. Lines 125 to 127. These numbers only add up to 161. What about the other 18 cities? 179 - 73 - 45 - 43 = 18. Am I misunderstanding this sentence?

Those 18 cities require more than 5 stores. To help clarify this (and add more information) we added the following:

“Jacksonville, FL and Miramar, FL are tied for the most stores required at 16.”

11. Can Figure 1 be moved up in the manuscript - closer to where you discuss the results in the text?

We moved Figure 1 up in the manuscript. The positioning of the figures will be at the journal’s discretion, as you know.

12. The units in the inset graph in Figure 1C are meters, right? Please indicate that.

Thanks for noticing this. It’s been updated.

13. I really like Figure 3!

Thanks!

14. Table 2. What does “average” indicate here? Average level of access? For instance, the average level of access for Seattle is 0km?

“Average” means the average level of access across all 500 cities (2290 m). The table contains the number of stores required to reach that level. Many cities, including Seattle, are already above the average level of access and so they don’t require any additional stores. To clarify, we changed the heading to that column (also suggested by another reviewer) and edited the caption:

“Average” indicates the average level of access across all 500 cities in the study (2290 m), so Indianapolis, ranked at 370, is first city in the list to require any additional supermarkets to reach the average level of access.

15. L170. Temporal variability of public transportation is also true of stores, no? Opening and closing times could also be incorporated? For more on this, look at
- Widener, M. J., & Shannon, J. (2014). When are food deserts? Integrating time into research on food accessibility. *Health & place*, 30, 1-3.

- Kaplan, K. H., Kirk, K. J., Lich, K. M., Palde, L. P. R., Van Allen, C., Nantz, E. L., ... & Knudsen, D. C. (2020). Accessibility to emergency food systems in south-central Indiana evaluated by spatiotemporal indices of pressure at county and pantry level. *Nature Food*, 1(5), 284-291.

This relates to the dimensions of accessibility, which we have improved our treatment of. Specifically, hours of operation are considered as part of the dimension on adequacy (also known as accommodation):

- **Introduction:** *“Following Penchansky and Thomas' framework \cite{Penchansky1981-qh}, as extended by Saurman \cite{Saurman2016-gj}, comprehensive food access depends on multiple factors: availability, geographic accessibility, adequacy or accommodation (e.g., store hours), affordability, acceptability (whether stores meet cultural preferences), and awareness. Our analysis focuses on geographic accessibility while acknowledging these broader dimensions.”*
- **Discussion:** *“Even within geographic accessibility, temporal patterns significantly shape access - store operating hours, variation in personal mobility throughout the day, and cyclical patterns of food purchasing tied to benefit distribution all affect when people can reach stores [Widener2014-en]. Research shows that people's food shopping patterns are influenced by multiple factors beyond simple proximity - including store quality, price, safety, and social networks [Shannon2016-wa].”*

16. L186. More is needed here. Cultural preferences, crime, etc. all shape who shapes where and when. Jerry Shannon's (Univ. of Georgia) work lays out a good deal of lit on why people shop where they do and when they do it.

- Shannon, J. (2016). Beyond the supermarket solution: Linking food deserts, neighborhood context, and everyday mobility. *Annals of the American Association of Geographers*, 106(1), 186-202.

This also ties to our improved inclusion of dimensions of access. Specifically, we have expanded our discussion: *“Research shows that people's food shopping patterns are influenced by multiple factors beyond simple proximity - including store quality, price, safety, and social networks \cite{Shannon2016-wa}. Residents often travel outside their immediate neighborhoods and visit multiple stores to balance these*

considerations with their cultural preferences and economic constraints \cite{Shannon2016-wa}.”

17. On L244, please indicate that distance is Manhattan, not Euclidean.

Please see the response to 18. below.

18. Line 251. This makes me think you are using Manhattan distance, which is great (better than Euclidean, in this instance). What data is the Open Source Routing Machine using for the routes, and where does it come from?

We are using network distance (not Euclidean or Manhattan). We added the following text to clarify our distance calculations:

“We calculated these distances using Open Source Routing Machine (OSRM) \cite{osrm} via the method described by Logan et al. \cite{Logan2019-fr}. Using data from Open Street Map \cite{OpenStreetMap}, OSRM calculates the shortest path along the road network from origin to destination. This method is more accurate than Euclidean or Manhattan distances because it is based on actual roadways, navigating around geographical barriers, such as freeways, waterways, and railroad tracks, as required.”

Reviewer #4 (Remarks on code availability):

The authors indicate that the method (optimization model) that they use in the paper comes from work in another manuscript; however, it seems that other manuscript has not been published to this point. I don't feel comfortable with my skills / knowledge to comment on whether or not their code / quantitative methods are up-to-par or at the level required. However, I would feel more confident commenting on this paper, in general, if the paper they're pulling the method from was published at this point.

Unfortunately, this paper is still under review.

Response to reviews - NCOMMS-24-27165-A

Dear reviewers,

Many thank you for your time and attention. We have responded to the comment and updated the checklist.

Thank you again.

Reviewer #1 (Remarks on code availability):

After reading the revised version of the paper, we think the authors have done excellent work and have addressed most of our previous comments in a reasonable way. We still have a question: The authors added that their analysis focuses on Census Block Groups within the selected cities' administrative boundaries with densities greater than 200 housing units per km². However, this does not seem to influence the results presented in Figure 4, which are identical in the original and the revised manuscript. Would it be possible to clarify why this is the case?

Many thanks to the reviewers. The reason that this figure hasn't changed between revisions is because that the earlier analysis was using these areas with a density of greater than 200 housing units per km² as well. The change made between revisions was that we clarified this in the methodology, rather than change the analysis.

Reviewer #2 (Remarks to the Author):

Reviewer #4 (Remarks to the Author):

Great job revising the text, both in terms of what the other reviewers and I mentioned, and in terms of general readability. Super work!